# Mutant *KRAS* promotes malignant pleural effusion formation

Theodora Agalioti[1,*], Anastasios D. Giannou[1,*], Anthi C. Krontira[1,*], Nikolaos I. Kanellakis[1,2], Danai Kati[1], Malamati Vreka[1,3], Mario Pepe[3], Magda Spella[1], Ioannis Lilis[1], Dimitra E. Zazara[1], Eirini Nikolouli[1], Nikolitsa Spiropoulou[1], Andreas Papadakis[1], Konstantina Papadia[4], Apostolos Voulgaridis[5], Vaggelis Harokopos[6], Panagiota Stamou[7], Silke Meiners[3], Oliver Eickelberg[3], Linda A. Snyder[8], Sophia G. Antimisiaris[4], Dimitrios Kardamakis[9], Ioannis Psallidas[1,2,**], Antonia Marazioti[1,**] & Georgios T. Stathopoulos[1,3,**]

Malignant pleural effusion (MPE) is the lethal consequence of various human cancers metastatic to the pleural cavity. However, the mechanisms responsible for the development of MPE are still obscure. Here we show that mutant *KRAS* is important for MPE induction in mice. Pleural disseminated, mutant *KRAS* bearing tumour cells upregulate and systemically release chemokine ligand 2 (CCL2) into the bloodstream to mobilize myeloid cells from the host bone marrow to the pleural space via the spleen. These cells promote MPE formation, as indicated by splenectomy and splenocyte restoration experiments. In addition, *KRAS* mutations are frequently detected in human MPE and cell lines isolated thereof, but are often lost during automated analyses, as indicated by manual versus automated examination of Sanger sequencing traces. Finally, the novel *KRAS* inhibitor deltarasin and a monoclonal antibody directed against CCL2 are equally effective against an experimental mouse model of MPE, a result that holds promise for future efficient therapies against the human condition.

[1] Laboratory for Molecular Respiratory Carcinogenesis, Department of Physiology, Faculty of Medicine, University of Patras, 26504 Rio, Greece. [2] Oxford Centre for Respiratory Medicine, Oxford University Hospitals NHS Trust, Churchill Hospital Old Road, Oxford OX3 7LE, UK. [3] Comprehensive Pneumology Center (CPC) and Institute for Lung Biology and Disease (iLBD), University Hospital, Ludwig-Maximilians University and Helmholtz Zentrum München, Member of the German Center for Lung Research (DZL), 81377 Munich, Germany. [4] Laboratory for Pharmaceutical Technology, Department of Pharmacy, School of Health Sciences, University of Patras, and Foundation for Research and Technology Hellas, Institute of Chemical Engineering, FORTH/ICE-HT, 26504 Rio, Greece. [5] Department of Pulmonary Medicine, Rio University Hospital, Faculty of Medicine, University of Patras, 26504 Rio, Greece. [6] Genomics Facility, Biomedical Sciences Research Center 'Alexander Fleming', Vari, Attica 16672, Greece. [7] Department of Hematology, Faculty of Medicine, University of Patras, Rio, Achaia 26504, Greece. [8] Oncology Discovery Research, Janssen R&D LLC, Spring House, Pennsylvania, 19477 USA. [9] Department of Radiation Oncology and Stereotactic Radiotherapy, Faculty of Medicine, University of Patras, 26504 Rio, Greece. * These authors contributed equally to this work. ** These authors jointly supervised this work. Correspondence and requests for materials should be addressed to G.T.S. (email: gstathop@upatras.gr).

The pleural cavities of two million cancer patients per year are affected by malignant pleural effusion (MPE), caused by primary malignant pleural mesothelioma or by metastatic cancers originating from the lung, breast, gastrointestinal tract or elsewhere[1]. MPE manifests with vascular leakiness that leads to fluid accumulation in the pleural space and is etiologically associated with fulminant inflammation and neovascularization, rather than mere tumour-induced lymphatic obstruction[2]. However, the reason why some patients with pleural tumours develop MPE while others do not remains unknown[3]. This dichotomous phenotype of 'wet' pleural carcinomatosis associated with a MPE versus 'dry' pleural carcinomatosis without a MPE is critical, since patients with even minimal effusions face a worse prognosis and limited treatment options[3,4]. Our previous work on experimental mouse models of MPE revealed that pleural tumour-secreted C–C motif chemokine ligand 2 (CCL2) mediates MPE formation by stimulating angiogenesis and vascular leakage and by driving myeloid cells, including monocytes and mast cells, from the bone marrow to the pleural metastatic milieu[5–7]. However, the molecular culprits responsible for tumour cell CCL2 secretion and subsequent MPE precipitation remain unknown.

EGFR, KRAS, PIK3CA, BRAF, MET, EML4/ALK, RET and other mutations have been identified in pleural tumour biopsies and pleural fluid aspirates from MPE patients[8–16]. EGFR mutations were recently implicated in MPE development and patients with KRAS-mutant lung adenocarcinomas were found to have more frequent pleural metastases compared with wild-type ones[17–19]. However, no study has addressed the role of KRAS mutations in MPE development.

We hypothesized that the ability of a tumour cell to induce a MPE once it homes to the pleural space is linked with an underlying molecular signature. To test this and to model the biologic events that follow pleural metastasis, we determined the mutation status of multiple murine and human cancer cell lines and simultaneously tested their ability to induce MPE by directly injecting them into the pleural space of appropriate recipient mice. Our results indicate that pleural homed cancer cells harboring activating KRAS mutations are competent of MPE induction. Moreover, we provide evidence that this genotype-phenotype link is primarily mediated via mutant KRAS-dependent CCL2 signalling that results in the recruitment of CD11b + Gr1 + myeloid cells to the pleural space, a phenomenon requiring intact splenic function. Importantly, we show that KRAS mutations are detectable in human MPE by careful analyses of Sanger sequencing traces and that mutant KRAS-mediated MPE is actionable.

## Results

**A link between KRAS mutations and MPE.** To identify a possible MPE-associated genotype, we cross-examined five murine C57BL/6-derived and five human cancer cell lines for genotype and MPE competence. For this, we directly injected $1.5 \times 10^5$ mouse or $10^6$ human tumour cells or $3 \times 10^6$ HEK293T benign human embryonic kidney cells into the pleural cavities of C57BL/6 (mouse cells) or NOD/SCID (human cells) mice. In parallel, we Sanger-sequenced the Kras, Egfr, Pik3ca and Braf transcripts of mouse cells after reverse-transcribing them to cDNAs and amplifying them with specific primers (Supplementary Table 1), and obtained mutation data for KRAS, EGFR, PIK3CA and BRAF genes of human cells from COSMIC[20]. KRAS mutations of human cells were also verified in-house. Among mouse cells, three Kras-mutant (Lewis lung carcinoma, LLC; MC38 colon adenocarcinoma; and AE17 malignant pleural mesothelioma, bearing heterozygous $Kras^{G12C}$, $Kras^{G13R}$, and $Kras^{G12C}$

mutations, respectively) and two Kras wild-type (B16F10 skin melanoma and PANO2 pancreatic adenocarcinoma) cell lines were identified, which were all free of additional mutations in Egfr, Pik3ca or Braf genes (Fig. 1a; Table 1). Among human cells, A549 lung adenocarcinoma cells and their derivatives, long-term passaged (LTP) A549 cells that have suffered Y chromosome loss, featured a heterozygous $KRAS^{G12S}$ mutation, while SKMEL2 skin melanoma, HT-29 colon adenocarcinoma, and HEK293T human embryonic kidney cells were KRAS wild-type (Table 1). These human cell lines also had wild-type EGFR, PIK3CA and BRAF genes, with the exception of HT-29 cells that harbor BRAF and PIK3CA mutations[20]. KRAS-mutant cell lines exhibited enhanced KRAS mRNA expression and RAS activity compared to KRAS wild-type cells (Supplementary Fig. 1a–d). Interestingly, upon pleural injection to appropriate hosts, all cell lines produced extensive pleural carcinomatosis, but exclusively KRAS-mutant cells gave rise to MPE (Fig. 1b–d; Table 1). To definitely test this in an isogenic cellular system, we derived lung adenocarcinoma cell lines from C57BL/6 and FVB mice. For this, C57BL/6 mice received ten and FVB mice four weekly intraperitoneal injections of the lung carcinogen urethane $(1 \, \mathrm{g \, kg^{-1}})$, as described elsewhere[21,22], and were killed after 10 months, followed by long-term lung tumour culture in vitro[23]. The resulting cell lines (C57BL/6 and FVB-derived urethane-induced lung adenocarcinoma, CULA and FULA cells, respectively) were tumourigenic when implanted subcutaneously in syngeneic mice. Importantly, three different FULA cell lines had three different Kras mutations (including Q61H, Q61R and G12V mutations), while CULA cells were Kras wild-type (Fig. 1a; Table 1). In accordance with the results from existing cell lines, all Kras-mutant FULA cell lines were MPE-competent, while Kras-wild-type CULA cells were not (Fig. 1b–d; Table 1). In summary, out of the 12 different cell lines tested, six out of six KRAS-mutant cell lines were MPE-competent and none out of six MPE-incompetent, while none out of six KRAS-wild-type cell lines was MPE-competent, and six out of six MPE-incompetent $(P = 0.0022; \chi^2\text{-test})$, indicating a statistically significant association between mutant KRAS and MPE.

**Myeloid cells in mutant KRAS-dependent MPE.** Kras-mutant tumour cell-triggered MPE was clinically important as mice with MPE succumbed significantly $(P < 0.0001; \text{log-rank test})$ earlier compared with mice with dry pleural carcinomatosis from Kras-wild-type cells (Supplementary Fig. 1e). In addition to early lethality, mutant Kras-dependent MPE development was associated with a massive influx of myeloid cells into the pleural space. This was investigated using irradiated C57BL/6 chimeras reconstituted with luminescent bone marrow from ubiquitously luminescent CAG.Luc.eGFP donor mice fully backcrossed to the C57BL/6 strain[6,7,24]. Fourteen days after pleural tumour cell injection, only chimeras injected with Kras-mutant tumour cells showed an increased thoracic bioluminescent signal (Supplementary Fig. 1f). This KRAS-dependent inflammatory response associated with MPE formation was predominated by both polymorphonuclear and mononuclear myeloid cells that expressed both CD11b and Gr1, and either Ly6c or Ly6g (Fig. 1e,f; Supplementary Figs 1g and 2a). MPE development triggered by KRAS-mutant cancer cells was associated with an influx of increased numbers of all kinds of myeloid cells into the pleural space, but not with the presence of newly-appearing morphologically or molecularly distinct cell types, since differential pleural cell counts and flow cytometry results were similar percentage-wise in mice with or without a MPE (Supplementary Fig. 2a). In addition to triggering a myeloid inflammatory response, KRAS-mutant pleural tumours and MPEs

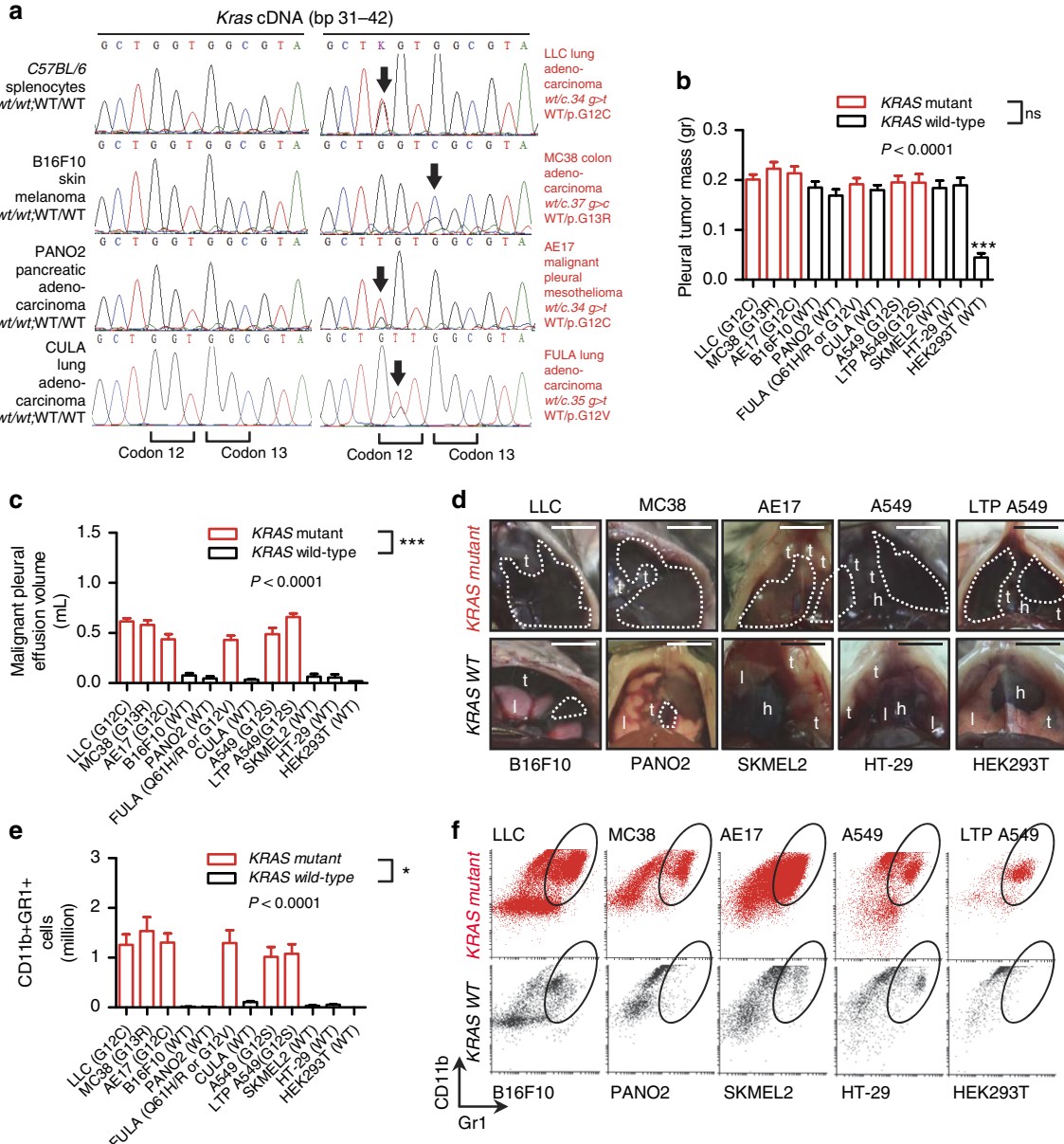

**Figure 1 | Selective induction of malignant pleural effusions by *KRAS*-mutant tumour cells.** Mutation status of and malignant pleural disease induction by twelve murine and human tumour cell lines after pleural delivery to appropriate recipient mice. (**a**) *Kras* cDNA Sanger sequencing traces of *C57BL/6* mouse splenocytes (control) and of five *C57BL/6* mouse tumour cell lines. Black arrows indicate heterozygous missense mutations in *Kras* codons 12 and 13. (**b**) Data summary of pleural tumor mass ($n = 53$, 26, 19, 30, 19, 27, 20, 16, 14, 14, 14, and 15, respectively, for LLC, MC38, AE17, B16F10, PANO2, FULA, CULA, A549, LTP A549, SKMEL2, HT-29, and HEK293T cells). (**c**) Data summary of malignant pleural effusion (MPE) volume ($n = 53$, 26, 19, 30, 19, 27, 20, 16, 14, 14, 14, and 15, respectively, for LLC, MC38, AE17, B16F10, PANO2, FULA, CULA, A549, LTP A549, SKMEL2, HT-29, and HEK293T cells). (**d**) Representative images of MPEs (dashed lines), pleural tumours (t), lungs (l), and hearts (h) imaged through the diaphragm. Scale bars, 1 cm. (**e**) Data summary of pleural CD11b + Gr1 + cells ($n = 5$–16 animals/group were analysed). (**f**) Representative dotplots and gating strategy for the quantification of pleural CD11b + Gr1 + cells. Data are presented as mean ± s.e.m. *P*, probability values for overall comparisons by one-way ANOVA. * and ***: $P < 0.05$ and $P < 0.001$ for the comparison between HEK293T cells and any other cell line (**b**) or for the comparison between any *Kras*-mutant and any *Kras*-wild-type cell line (**c,e**) by Bonferroni post-tests. *WT*, wild-type; LLC, *C57BL/6* Lewis lung carcinoma; MC38, *C57BL/6* colon adenocarcinoma; AE17, *C57BL/6* malignant pleural mesothelioma; B16F10, *C57BL/6* malignant skin melanoma; PANO2, *C57BL/6* pancreatic adenocarcinoma; FULA, *FVB* urethane-induced lung adenocarcinoma; CULA, *C57BL/6* urethane-induced lung adenocarcinoma; A549, human lung adenocarcinoma; LTP A549, long-term passaged A549 cells having lost the Y chromosome; SKMEL2, human malignant skin melanoma; HT-29, human colon adenocarcinoma; HEK293T, human embryonic kidney cells.

showed enhanced angiogenic and vasoactive potential in several *in vivo* assay systems compared with *KRAS*-wild-type tumours (Supplementary Fig. 2b–d). However, we did not detect an increased proliferative or clonogenic capacity specifically characterizing *KRAS*-mutant cells and tumours compared with *KRAS*-wild-type ones (Supplementary Fig. 2e–i). This was in accord with the equal total mass of pleural tumors per mouse observed across pleural-injected *KRAS*-mutant and wild-type tumour cells (Fig. 1b). Collectively, these results suggested that mutant *KRAS*-driven MPE is associated with induction of an inflammatory, angiogenic, and vasoactive response in the pleural space, but not necessarily with enhanced pleural tumour growth.

| Table 1 | Incidence of murine malignant pleural effusions and mutation status of twelve tumour cell lines. | | | | | | | | |
|---|---|---|---|---|---|---|---|---|---|
| n = 265 | MPE | | | KRAS | EGFR | BRAF | PIK3CA | P value | |
| | No | Yes | % | | | | | | |
| LLC | 1 | 52 | 98 | G12C | WT | WT | WT | 0.0537 | |
| MC38 | 1 | 25 | 96 | G13R | WT | WT | WT | 0.2954 | |
| AE17 | 3 | 16 | 84 | G12C | WT | WT | WT | 1.0000 | |
| B16F10 | 24 | 6 | 20 | WT | WT | WT | WT | 0.000012 | |
| PANO2 | 18 | 1 | 5 | WT | WT | WT | WT | 0.00000064 | |
| FULA | 3 | 24 | 89 | G12V/Q61R/H | WT | WT | WT | 0.6796 | |
| CULA | 18 | 2 | 10 | WT | WT | WT | WT | 0.000003 | |
| A549 | 2 | 14 | 88 | G12S | WT | WT | WT | 1.0000 | |
| LTP A549 | 0 | 14 | 100 | G12S | WT | WT | WT | 0.2443 | |
| SKMEL2 | 13 | 1 | 7 | WT | WT | WT | WT | 0.000012 | |
| HT-29 | 13 | 1 | 7 | WT | WT | V600E/T119S | P449T | 0.000012 | |
| HEK293T | 15 | 0 | 0 | WT | WT | WT | WT | 0.00000044 | |

AE17, *C57BL/6* malignant pleural mesothelioma; A549, human lung adenocarcinoma; B16F10, *C57BL/6* malignant skin melanoma; CULA, *C57BL/6* urethane-induced lung adenocarcinoma; FULA, *FVB* urethane-induced lung adenocarcinoma; HEK293T, human embryonic kidney cells; LLC, *C57BL/6* Lewis lung carcinoma; LTP A549, long-term passaged A549 cells having lost the Y chromosome; MC38, *C57BL/6* colon adenocarcinoma; PANO2, *C57BL/6* pancreatic adenocarcinoma; SKMEL2, human malignant skin melanoma; WT, wild-type. Shown is number of mice (*n*) that developed dry pleural carcinomatosis (no MPE; <100 µl pleural fluid) and number (*n*) and percentage (%) of mice that developed MPE (≥100 µl pleural fluid). $P < 0.0001$ for overall comparison by $\chi^2$-test. *P*, probability values for comparison with AE17 cells, the *KRAS*-mutant cell line with the lowest MPE incidence by Fischer's exact tests.

**Mutant *KRAS* promotes MPE.** To corroborate the link between *KRAS* mutations and MPE, we undertook both shRNA-mediated *KRAS* silencing in cell lines harboring mutant *KRAS*, as well as mutant *KRAS* overexpression in cell lines harboring wild-type *KRAS*. Stable transduction of six different *Kras*-mutant mouse tumour cells with lentiviral-delivered *Kras*-specific shRNA (sh*Kras*) resulted in diminished expression of both murine *Kras* isoforms (2A and 2B) and decreased RAS signalling compared with random (control, sh*C*) shRNA, whereas overexpression of mutant *Kras*[G12C] isoforms in murine and human cell lines carrying wild-type *KRAS* via retroviral transduction enhanced the respective KRAS protein levels and increased RAS signalling (Supplementary Fig. 3). Manipulation of *KRAS* signalling did not result in obvious enhancements of tumour cell proliferation or survival *in vitro*; on the contrary, overexpression of *Kras*[G12C]2A in PANO2 cells and of *Kras*[G12C]2B in B16F10 cells slowed their growth rate (Supplementary Fig. 4). However, upon direct inoculation of all parental and daughter *KRAS*-modulated cell lines into the pleural space of appropriate (*C57BL/6*, *FVB* or *NOD/SCID*) host mice, all mice developed similar extent of pleural carcinomatosis, but expression of mutant *KRAS* was a cardinal determinant of MPE in all cell lines examined (Fig. 2a,b; Table 2; Supplementary Fig. 5). More specifically, *Kras* silencing universally abrogated MPE formation by LLC, MC38, AE17 and three different FULA cell lines, that is, in cells harbouring either *KRAS* G12C (LLC & AE17 cells), G13R (MC38 cells), G12V, Q61H or Q61R (FULA cells) mutations, whereas oncogenic *Kras*[G12C] expression conferred MPE competence to B16F10, PANO2, SKMEL2, and HEK293T cells. Remarkably: (i) expression of *Kras*[G12C] isoform 2A conferred enhanced MPE competence to PANO2 cells compared with *Kras*[G12C] isoform 2B, although the later was more abundantly expressed by *KRAS*-mutant cancer cells; and (ii) mutant *KRAS* expression converted even benign HEK293T cells to MPE competence (Fig. 2a,b; Table 2; Supplementary Fig. 5). Using pleural injection of parental and *KRAS*-modulated MC38 and PANO2 cells into *C57BL/6* chimeras reconstituted with *CAG.Luc.eGFP* bone marrow, we identified that mutant *KRAS* is not only responsible for MPE development, but also for the associated pleural influx of CD11b + Gr1 + cells (Fig. 2c; Supplementary Fig. 5c,d). Taken together, these results indicated that mutant *KRAS* is dispensable for pleural tumour growth, but important for MPE development and for the associated systemic recruitment of CD11b + Gr1 + myeloid cells, and suggested that *KRAS* must be responsible for the secretion of a solute mediator of MPE by tumour cells.

**KRAS-mutant tumour cells signal via CCL2 to host cells.** To identify the MPE mediator(s) downstream of mutant *KRAS* and to tease out the transcriptional signature of mutant *KRAS* on tumour cells, we performed comparative microarray-based transcriptome profiling of *Kras*-mutant and wild-type mouse tumour cells versus benign airway cells. Unsupervised clustering according to global gene expression revealed that *Kras*-mutant cell lines clustered closely together (Supplementary Fig. 6a). Individual gene analysis identified 25 transcripts overrepresented more than 10-fold in *KRAS*-mutant, but not in *KRAS*-wild-type, cell lines compared with benign cells (Fig. 3a; Table 3). Microarray results were verified by qPCR and ELISA (Fig. 3b; Supplementary Figs 6b,c). Furthermore: (i) manipulation of mutant *KRAS* expression resulted in parallel changes in *Ccl2* expression; (ii) cell culture media conditioned by *KRAS*-mutant tumour cells featured markedly elevated CCL2 levels compared with media conditioned by *KRAS*-wild-type tumour cells; and (iii) mice bearing in their pleural space *KRAS*-mutant tumour cells featured markedly elevated serum CCL2 levels compared with mice harbouring *KRAS*-wild-type tumour cells (Supplementary Fig. 6d–g). To corroborate CCL2 as the down-stream effector of mutant *KRAS* that mediates MPE *in vivo*, we directly delivered LLC (*Kras*[G12C]), MC38 (*Kras*[G13R]), and PANO2 cells overexpressing *Kras*[G12C] isoform 2A into the pleural space of *Ccr2*-gene-deficient mice (*Ccr2 − /−* ; the gene encoding the cognate receptor of CCL2)[25] and *C57BL/6* controls. In accord with our hypothesis, *Ccr2 − /−* mice were protected against MPE induced by all three *Kras*-mutant tumour cell lines and displayed reduced CCR2 expression by pleural fluid cells and decreased accumulation of CD11b + Gr1 + cells in the pleural space (Fig. 3c; Supplementary Fig. 6h,i). Collectively, these data suggest that mutant *KRAS* drives MPE development via systemic CCL2 signalling to CCR2 + host cells.

**Mutant KRAS recruits splenic CD11b + Gr1 + cells to MPE.** We next sought to identify the systemic recruitment patterns of myeloid cells during MPE development. For this, *C57BL/6* chimeras reconstituted with *CAG.Luc.eGFP* bone marrow were inoculated with *Kras*-mutant pleural tumour cells and were serially imaged for bioluminescence. Although immediately after pleural tumour cell delivery the myeloid-emitted bioluminescent signal was primarily identified over the hematopoietic bones, it sequentially translocated to the upper left abdomen (days 10–12 post-tumour cell injection) before appearing in the thorax at days

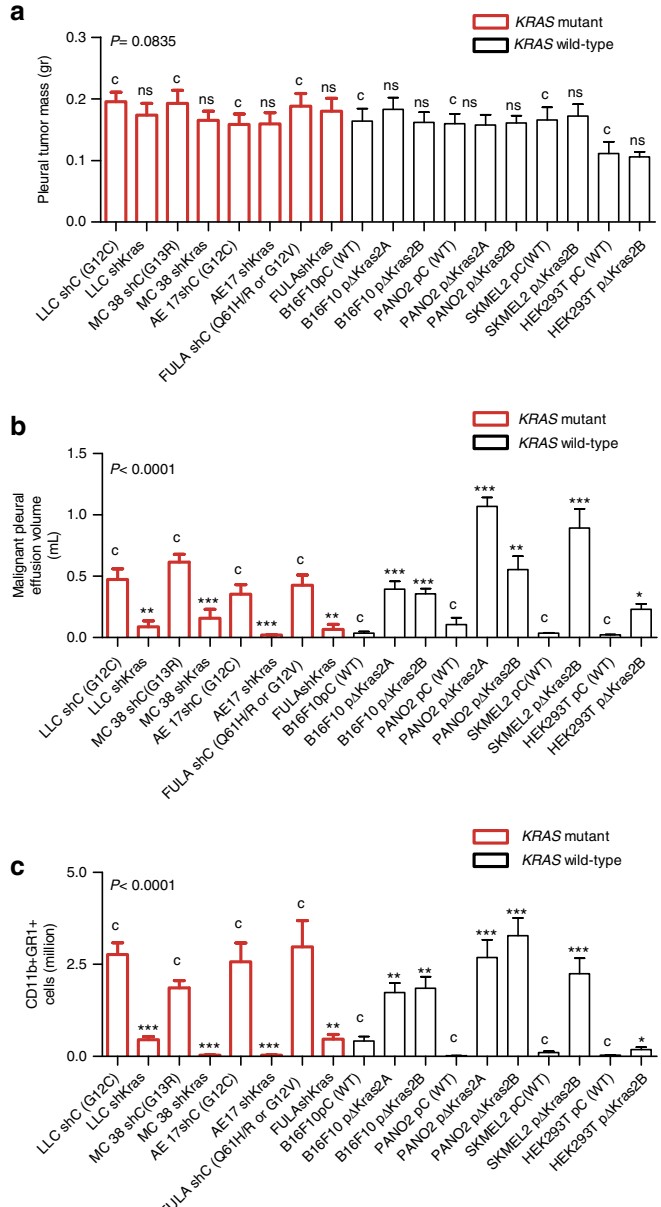

**Figure 2 | Mutant *KRAS* promotes malignant pleural effusion development.** Impact of shRNA-mediated *Kras* silencing on MPE competence of cell lines harboring mutant *Kras*, and of mutant *Kras*G12C overexpression in cell lines harboring wild-type *KRAS*. (**a**) Data summary of pleural tumor mass ($n = 14$, 12, 11, 14, 11, 11, 9, 9, 11, 10, 13, 16, 12, 16, 9, 10, 9, and 9, respectively, for LLC sh*C*, LLC sh*Kras*, MC38 sh*C*, MC38 sh*Kras*, AE17 sh*C*, AE17 sh*Kras*, FULA sh*C*, FULA sh*Kras*, B16F10 p*C*, B16F10 p*ΔKras*2A, B16F10 p*ΔKras*2B, PANO2 p*C*, PANO2 p*ΔKras*2A, PANO2 p*ΔKras*2B, SKMEL2 p*C*, SKMEL2 p*ΔKras*2B, HEK293T p*C*, and HEK293T p*ΔKras*2B cells). (**b**) Data summary of MPE volume ($n = 14$, 12, 11, 14, 11, 11, 9, 9, 11, 10, 13, 16, 12, 16, 9, 10, 9, and 9, respectively, for LLC sh*C*, LLC sh*Kras*, MC38 sh*C*, MC38 sh*Kras*, AE17 sh*C*, AE17 sh*Kras*, FULA sh*C*, FULA sh*Kras*, B16F10 p*C*, B16F10 p*ΔKras*2A, B16F10 p*ΔKras*2B, PANO2 p*C*, PANO2 p*ΔKras*2A, PANO2 p*ΔKras*2B, SKMEL2 p*C*, SKMEL2 p*ΔKras*2B, HEK293T p*C*, and HEK293T p*ΔKras*2B cells). (**c**) Data summary of pleural CD11b + Gr1 + cells ($n = 7$–11/group were analysed). Data are presented as mean ± s.e.m. *P*, probability values for overall comparisons by one-way ANOVA. ns, *, **, and ***: $P > 0.05$, $P < 0.05$, $P < 0.01$, and $P < 0.001$ for the comparison between the indicated cell line and the respective control (**c**) by Student's *t*-test or one-way ANOVA with Bonferroni post-tests, as appropriate. *WT*, wild-type; sh*C*, random shRNA; sh*Kras*, anti-*Kras*-specific shRNA; p*C*, control (empty) overexpression vector; p*ΔKras*2A and p*ΔKras*2B, overexpression vectors encoding mutant mouse *Kras*G12C isoforms A and B, respectively; *WT*, wild-type; LLC, *C57BL/6* Lewis lung carcinoma; MC38, *C57BL/6* colon adenocarcinoma; AE17, *C57BL/6* malignant pleural mesothelioma; B16F10, *C57BL/6* malignant skin melanoma; PANO2, *C57BL/6* pancreatic adenocarcinoma; FULA, *FVB* urethane-induced lung adenocarcinoma; SKMEL2, human malignant skin melanoma; HEK293T, human embryonic kidney cells.

12–14 post tumour cell injection (Supplementary Fig. 7a). Splenectomy abolished this abdominal myeloid-borne signal that was recapitulated from explanted spleens (Fig. 3d). In addition, CCR2 + CD68 + myeloid cells were identified in the splenic marginal zones and the pleural cavities of mice with MPE induced by *KRAS*-mutant cells, but not of naive mice

(Supplementary Fig. 7b). These results suggested that CD11b + Gr1 + myeloid cells are mobilized by mutant *KRAS*-driven CCL2-mediated signalling from the bone marrow to MPE via the spleen. On the basis of this evidence and the existing literature[26–28], we hypothesized that the splenic passage of CD11b + Gr1 + cells is essential for MPE formation. To test

**Table 2 | Incidence of malignant pleural effusions caused by parental and *KRAS* -modulated tumour cell lines.**

| $n = 206$ | MPE | | | P value |
|---|---|---|---|---|
| | **no** | **yes** | **%** | |
| LLC sh*C* (G12C) | 3 | 11 | 79 | c |
| LLC sh*Kras* | 10 | 2 | 17 | 0.0048 |
| MC38 sh*C* (G13R) | 0 | 11 | 100 | C |
| MC38 sh*Kras* | 10 | 4 | 29 | 0.0005 |
| AE17 sh*C* (G12C) | 1 | 10 | 91 | c |
| AE17 sh*Kras* | 11 | 0 | 0 | <0.0001 |
| FULA sh*C* (G12V, Q61R/H) | 1 | 8 | 89 | c |
| FULA sh*Kras* | 8 | 1 | 11 | 0.0034 |
| B16F10 p*C* (WT) | 10 | 1 | 9 | c |
| B16F10 pΔ*Kras*2A | 0 | 10 | 100 | <0.0001 |
| B16F10 pΔ*Kras*2B | 1 | 12 | 92 | <0.0001 |
| PANO2 p*C* (WT) | 14 | 2 | 13 | c |
| PANO2 pΔ*Kras*2A | 0 | 12 | 100 | <0.0001 |
| PANO2 pΔ*Kras*2B | 4 | 12 | 75 | 0.0010 |
| SKMEL2 p*C* (WT) | 9 | 0 | 0 | c |
| SKMEL2 pΔ*Kras*2B | 2 | 8 | 80 | 0.0007 |
| HEK293T p*C* (WT) | 9 | 0 | 0 | c |
| HEK293T pΔ*Kras*2B | 2 | 7 | 78 | 0.0023 |

AE17, *C57BL/6* malignant pleural mesothelioma; B16F10, *C57BL/6* malignant skin melanoma; FULA, *FVB* urethane-induced lung adenocarcinoma; HEK293T, human embryonic kidney cells; LLC, *C57BL/6* Lewis lung carcinoma; MC38, *C57BL/6* colon adenocarcinoma; PANO2, *C57BL/6* pancreatic adenocarcinoma; SKMEL2, human malignant skin melanoma; WT, wild-type.
Shown is number of mice (*n*) that developed dry pleural carcinomatosis (no MPE; <100 μl pleural fluid) and number (*n*) and percentage (%) of mice that developed MPE (≥100 μl pleural fluid).
$P < 0.0001$ for overall comparison by $\chi^2$ test. *P*, probability values for comparison with parental control cells (c) by Fischer's exact tests.

this, we delivered MC38 cells (*Kras*[G13R]) or PANO2 cells expressing mutant *Kras*[G12C] isoform 2A or 2B to the pleural cavities of splenectomized and sham-operated *C57BL/6* mice after allowing two weeks for recovery. Indeed, splenectomy markedly protected *C57BL/6* mice from MPE, prolonged their survival, and prevented pleural accumulation of CD11b + Gr1 + myeloid cells (Fig. 3e; Supplementary Fig. 7c,d). Similarly, splenectomy protected *NOD/SCID* mice from A549-induced MPE (*KRAS*[G12S]; Supplementary Fig. 7e), further suggesting that myeloid and not lymphoid splenic cells promote MPE in these lymphoid-deficient mice. Splenectomy-conferred protection was long-lived, as even mice collected 30 days post-tumour cell injection did not have MPE (Supplementary Fig. 7c). To address whether splenic CD11b + Gr1 + cells are required for MPE development, tumour-naive and tumour-bearing *CAG.Luc.eGFP* mice were used as splenocyte donors to splenectomized pleural MC38 (*Kras*[G13R])-bearing *C57BL/6* mice. These *CAG.Luc.eGFP* donors received pleural injections of saline (naive splenocyte), control shRNA-expressing MC38 cells (MC38 sh*C*-educated splenocyte) or *Kras* specific-shRNA expressing MC38 cells (MC38 sh*Kras*-educated splenocyte) and 13 days later, their spleens were collected and processed to single-cell suspensions. In parallel, splenectomized or sham-operated *C57BL/6* hosts received pleural MC38 cell injections. At post-injection day 9, splenectomized animals received five million intravenous splenocytes obtained from naive, sh*C* or sh*Kras* MC38-bearing donors whereas, at post-injection day 13, these mice were analysed for MPE incidence, volume, survival, and for Luc + CD11b + Gr1 + recruited pleural cells (derived from transplanted splenocytes). Interestingly, only splenocytes from donors inoculated with MC38 cells bearing intact mutant *KRAS* signalling were able to translocate to the pleura and promote MPE formation in splenectomized mice harbouring pleural MC38 cells (Supplementary Fig. 7f–h). Taken together, these results indicated that *KRAS*-mutant pleural tumours induce the sequential recruitment of CD11b + Gr1 + cells from the bone marrow to the spleen and into the pleural cavity. Furthermore, that during MPE formation, bone marrow-borne, splenic CD11b + Gr1 + cells are conditioned by solute mediators

secreted by *KRAS*-mutant pleural tumours (possibly CCL2) and functionally contribute to MPE development.

**KRAS mutations in human MPE**. We next Sanger-sequenced the *KRAS* transcripts of 12 human MPEs caused by metastatic lung adenocarcinomas according to established protocols[16]. Interestingly, *KRAS* mutations were present in numerous MPEs, but were not always readily detectable by automated Sanger sequencing trace analysis using BioEdit software[29], since mutant base traces were often hidden underneath wild-type traces superimposed by the other *KRAS* allele, or by tumour-infiltrating stromal cells (Fig. 4a,b; Table 4). We also analysed recently published data of the site of recurrence of 481 resected non-small cell lung cancers according to *KRAS* and *EGFR* mutation status, and found that *KRAS* mutations overall were highly significantly ($P < 0.0001$; Fischer's exact test) associated with pleural recurrence (Table 4). We went on to derive cell lines from eight patients with lung adenocarcinoma-induced MPE that were initially tested *KRAS* wild-type. Interestingly, *KRAS* mutations were frequently identified in MPE cell lines initially tested wild-type (Fig. 4c–e; Table 4). These results suggested that: (i) *KRAS* mutations are present in a substantial proportion of patients with lung adenocarcinoma-caused MPE in Europe; (ii) *KRAS* mutation frequency may be underestimated in MPE samples analysed automatically; and (iii) our observations in mice may also hold true in humans.

**Targeting KRAS is effective against MPE development**. To determine the potential efficacy of KRAS inhibition against MPE, the novel KRAS inhibitor deltarasin[30] was administered daily intraperitoneally at 15 mg kg[−1], side-by-side with a saline control treatment, to mice with established pleural tumours. For this, treatments commenced at day 4–14 post-mouse tumour cell injections and at day 14 post-human tumour cell injections to allow initial tumour implantation in the pleural space[6,7]. At day 13 after pleural injection of MC38 cells (*Kras*[G13R]), deltarasin-treated *C57BL/6* mice developed fewer and smaller MPEs,

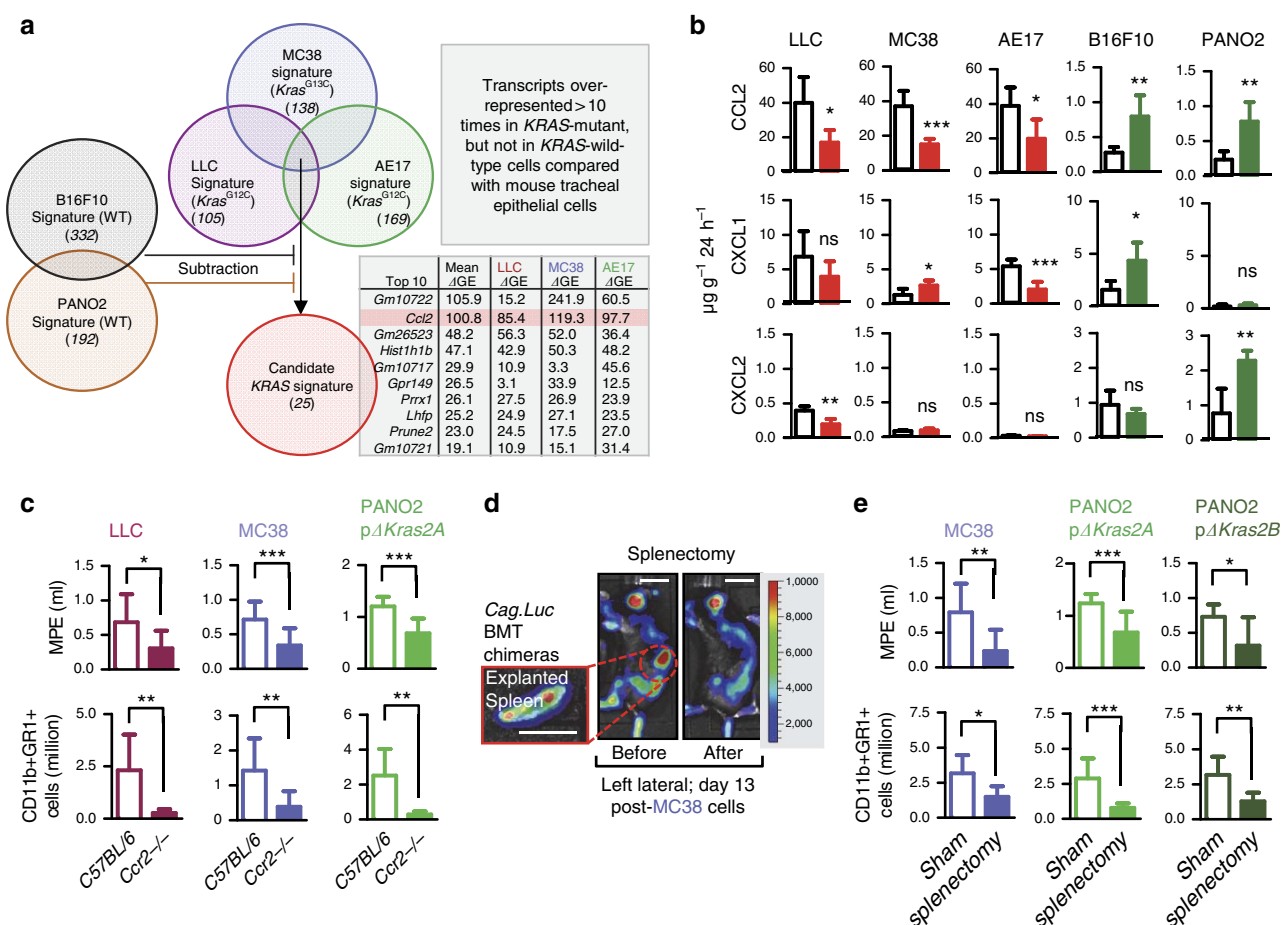

**Figure 3 | Mutant KRAS signals via CCL2 to recruit splenic myeloid cells to malignant pleural effusions.** (**a**) Comparative transcriptome analysis of mouse tumour cell lines with defined *Kras* mutation status versus benign airway epithelial cells by microarray. Diagram depicting the analytic strategy employed to identify the transcriptional signature of mutant *Kras* comprised of 25 genes (top ten shown in table), among which *Ccl2* ranked second. (**b**) Chemokine protein secretion by parental (white bars: cells stably expressing random shRNA or control overexpression vector) and *Kras*-modulated (red bars: cells stably expressing anti-*Kras*-specific shRNA; green bars: cells stably expressing vector encoding mutant mouse *Kras*[G12C] isoform B) murine cell lines by ELISA showing transcriptional regulation of CCL2, but not of CXCL1 and CXCL2, by mutant *Kras* (n = 5–7/group). (**c**) Data summaries of malignant pleural effusion (MPE) volume (top; LLC: n = 9/group; MC38: n = 14–15/group; PANO2 pΔKras2A: n = 8–18/group) and pleural CD11b + Gr1 + cells (bottom; LLC: n = 9/group; MC38: n = 14–15/group; PANO2 pΔKras2A: n = 5/group) of *Ccr2 − / −* and *C57BL/6* control mice after intrapleural injection of three different tumour cell lines. pΔKras2A, vector encoding mouse *Kras*[G12C] isoform A. (**d**) Representative bioluminescent images of chimeric *C57BL/6* mouse transplanted with bioluminescent bone marrow from *CAG.Luc.eGFP* donor before and after splenectomy performed at day 13 after intrapleural MC38 cells. Scale bars, 1 cm. (**e**) Data summaries of MPE volume (top; n = 9/group) and pleural CD11b + Gr1 + cells (bottom; n = 9/group) of *C57BL/6* mice pretreated with sham surgery or splenectomy followed by intrapleural injection of MC38 cells, or PANO2 cells expressing pΔKras2A or pΔKras2B two weeks later. Data are presented as mean ± s.d. ns, *, **, and ***: P > 0.05, P < 0.05, P < 0.01, and P < 0.001 for comparison with parental lines (**b**), between the two mouse strains (**c**), or between different surgeries (**e**) by Student's t-test.

retarded pleural tumour dissemination and decreased pleural CD11b + Gr1 + accumulation compared with controls (Fig. 5a; Table 5). Furthermore, *in vitro* treatment of MC38 cells with deltarasin resulted in almost complete elimination of CCL2 secretion (Fig. 5b). To test the impact of KRAS blockade in a more human-relevant setting, *NOD/SCID* mice received deltarasin and control treatments starting at two weeks after pleural delivery of one million A549 cells (*KRAS*[G12S]). At day 30 after tumour cell injection, deltarasin-treated mice had markedly decreased MPE volume and incidence compared with controls (Fig. 5c; Table 5). We also explored direct intrapleural targeted deltarasin delivery against experimental MPE, since chronic KRAS inhibition may result in marked toxicity. For this, *C57BL/6* mice received pleural MC38 cells, followed by a single intrapleural injection of liposomal-encapsulated deltarasin (15 mg kg−1; one single dose equal to the daily intraperitoneal drug dose

administration) or empty liposomes[31,32] on day seven post-tumour cells. Interestingly, single-dose intrapleural liposomal deltarasin exhibited equal efficacy with repetitive intaperitoneal drug treatment, halting both MPE accumulation and CD11b + Gr1 + cell influx (Fig. 5d; Table 5). We finally cross-examined the effects of deltarasin and of a well-characterized neutralizing anti-CCL2 antibody[6,7,33,34]. For this, *C57BL/6* mice received intrapleural PANO2 cells stably expressing *Kras*[G12C] isoforms 2A or 2B. After 4 or 14 days, respectively, mice started receiving daily intraperitoneal deltarasin (15 mg kg−1) or anti-CCL2 antibody (50 mg kg−1) every 3 days. Control mice received daily saline injections and IgG2a control antibody (50 mg kg−1) every 3 days. Interestingly, both treatments were equally effective in reducing MPE incidence and volume, as well as CD11b + Gr1 + cell accumulation (Fig. 5e; Table 5). These results indicated that deltarasin is effective in halting MPE

**Table 3 | Candidate mutant *Kras* transcriptome signature.**

| Gene symbol | Gene name | ΔGE LLC* | ΔGE MC38* | ΔGE AE17* |
|---|---|---|---|---|
| *Asns* | Asparagine synthetase | +12.7 | +11.6 | +11.2 |
| *Bcat1* | Branched chain aminotransferase 1, cytosolic | +11.2 | +17.6 | +12.2 |
| *Casp3* | Caspase 3 | +12.7 | +16.0 | +17.4 |
| *Ccl2* | Chemokine (C–C motif) ligand 2 | +85.4 | +119.3 | +97.7 |
| *Ccl7* | Chemokine (C–C motif) ligand 7 | +11.3 | +14.4 | +17.8 |
| *Cep170* | Centrosomal protein 170 | +15.3 | +11.6 | +13.5 |
| *Dab2* | Disabled 2, mitogen-responsive phosphoprotein | +12.7 | +18.2 | +10.4 |
| *Dusp9* | Dual specificity phosphatase 9 | +13.1 | +15.5 | +17.0 |
| *Gm10717* | Predicted gene 10717 | +10.9 | +33.3 | +45.6 |
| *Gm10721* | Predicted gene 10721 | +10.9 | +15.1 | +31.4 |
| *Gm10722* | Predicted gene 10722 | +15.2 | +241.9 | +60.5 |
| *Gm26523* | Predicted gene, 26523 | +56.3 | +52.0 | +36.4 |
| *Gpr149* | G protein-coupled receptor 149 | +33.1 | +33.9 | +12.5 |
| *Hist1h1b* | Histone cluster 1, H1b | +42.9 | +50.3 | +48.2 |
| *Hjurp* | Holliday junction recognition protein | +10.3 | +14.4 | +11.3 |
| *Lhfp* | Lipoma HMGIC fusion partner | +24.9 | +27.2 | +23.5 |
| *Mpp1* | Membrane protein, palmitoylated | +10.6 | +16.3 | +21.3 |
| *Nid1* | Nidogen 1 | +13.6 | +16.0 | +15.0 |
| *Nid1* | Nidogen 1 | +10.3 | +15.2 | +11.3 |
| *Prrx1* | Paired related homeobox 1 | +27.5 | +26.9 | +23.9 |
| *Prune2* | Prune homolog 2 (*Drosophila*) | +24.5 | +17.5 | +27.0 |
| *Psat1* | Phosphoserine aminotransferase 1 | +12.3 | +13.3 | +11.1 |
| *S100a4* | S100 calcium binding protein A4 | +13.4 | +12.5 | +10.1 |
| *Slc30a4* | Solute carrier family 30 (Zinc transporter), member 4 | +20.6 | +11.7 | +16.2 |
| *Snora17* | Small nucleolar RNA, H/ACA box 17 | +18.3 | +11.5 | +26.6 |

The 25 transcripts identified from comparative analyses of global gene expression between five murine cancer cell lines (LLC, MC38, AE17, B16F10, and PANO2 cells) and benign airway epithelial cells by alphabetic order.
*ΔGE, fold-difference in gene expression between *Kras*-mutant tumour cells and benign airway epithelial cells. *Ccl2* was the most consistently over-represented transcript across all three cell lines examined. LLC, *C57BL/6* Lewis lung carcinoma; MC38, *C57BL/6* colon adenocarcinoma; AE17, *C57BL/6* malignant pleural mesothelioma.

induction by *KRAS*-mutant tumour cells and suggested that mutant *KRAS*-driven MPE in humans may also be actionable.

## Discussion

The dichotomous phenotype of primary and metastatic pleural tumours, some of which are associated with an MPE whereas others are not, is of paramount clinical importance, and prompted us to hypothesize that a causative molecular signature underlines MPE formation[2–4]. We show that cancer cells bearing different *KRAS* mutations cause MPE upon pleural dissemination and that mutant *KRAS* is important for experimental MPE development. Furthermore, that mutant *KRAS*-driven MPE is attributed to a CCL2-dependent signalling cascade that is necessary for the sequential translocation of CD11b + Gr1 + cells from the bone marrow to the spleen and the tumour-involved pleural cavity, where, in turn, these cells promote MPE formation. Proof-of-principle clinical data indicate that *KRAS* mutations are present in a substantial proportion of MPE patients in Europe and that they might be underestimated by automated sequencing analyses. Finally, we show that pharmacologic interception of this newly identified *KRAS*-driven, CCL2-mediated pathway to MPE can prevent MPE development.

The newly identified genotype-disease connection between mutant *KRAS* and MPE was corroborated using 12 different isogenic cellular systems of parental and daughter *KRAS*-modulated cell lines. In each and single one of these systems, mutant *KRAS* was required and sufficient for MPE. Particularly impressive was the switch of PANO2 cells upon *Kras*[G12C]2A expression from complete MPE incompetence during a month's observation to acute and lethal MPE induction within 7 days. The same is true for benign HEK293T that were rendered MPE-proficient by isolated expression of *Kras*[G12C]2B *per se*. But is the proposed mutant *KRAS*-MPE link clinically relevant? *KRAS*

mutations are not frequently found in human MPE as opposed to *EGFR* and *EML4/ALK* mutations[8–16]. First, we believe that *KRAS* mutations are not looked for because they are considered not actionable and mutually exclusive to *EGFR* mutations, notions that are currently being revisited[8–16,35,36]. We show the clinical data that indicate that *KRAS* mutations are frequent in European patients with MPE from lung adenocarcinoma and that they might be underappreciated. A recent study also showed how *KRAS* mutations can be missed in MPE samples, but persist in cultured cell lines derived from the same patients[16], a finding recapitulated in patients from our centre. To this end, most MPE sequencing studies were performed in Asian populations with high *EGFR* and *EML/ALK* and low *KRAS* mutation frequencies[8–16,35,36]. A recent study of European patients with resected lung tumours clearly showed that *KRAS* mutations are linked with pleural spread[19]. Third, pleural tumours are diffuse-multifocal and probably multiclonal[36] and it is conceivable that mutant *KRAS* MPE-initiating cells escape detection in focal pleural tumour tissue biopsies and low-volume pleural fluid aspirates.

We do not claim that *KRAS* mutations are the only ones that cause MPE in humans and postulate mutant *KRAS* effects to be class effects shared by all driver mutations aligned along the KRAS pathway, including *EGFR*, *KRAS*, *PIK3CA*, *BRAF*, *MET*, *EML4/ALK*, *RET* and others. To this end, mutant *EGFR* was recently shown to cause MPE when expressed in H1299 human lung adenocarcinoma cells[17]. However, possible pathogenic roles for other lung cancer drivers in MPE remain to be shown. Together with the advent of MPE sequencing techniques[8–16], such developments could lead to targeted therapies for MPE in the near future. Moreover, MPE is a clinically heterogeneous set of diseases from a number of primary sites. *KRAS* mutations are more relevant to patients with lung, pancreatic, and colon cancers and leukaemias. In other tumors (that is, breast cancer) other

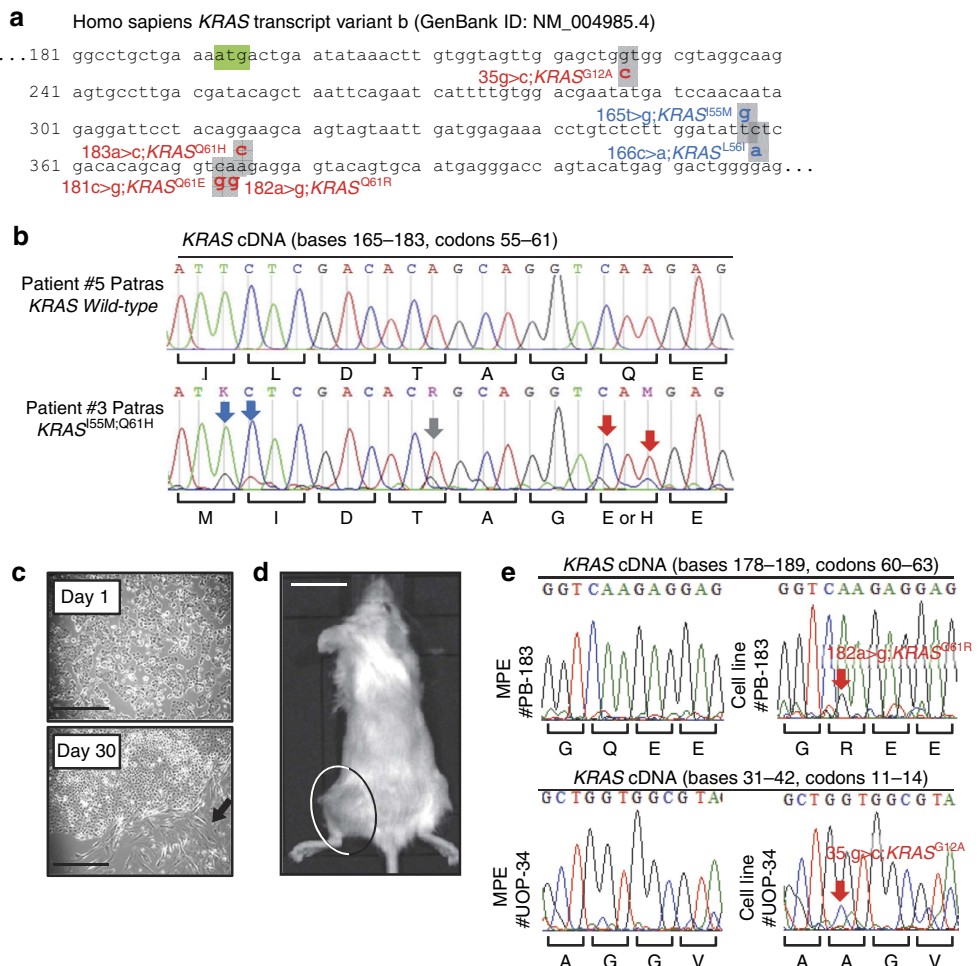

**Figure 4 | *KRAS* mutations in human malignant pleural effusions.** (**a–c**) Sanger sequencing results of human malignant pleural effusions (MPE) caused by metastatic lung adenocarcinomas from Institution 1. (**a**) Partial sequence of *Homo Sapiens KRAS* isoform b transcript showing start codon (green box) and missense mutations identified (grey boxes and callouts). Red and blue fonts indicate, respectively, known pathogenic mutations and mutations of unknown significance based on COSMIC[20]. (**b**) Partial Sanger-sequencing traces from two patients showing corresponding sequences of patient with wild-type *KRAS* alleles and of another with four different *KRAS* mutations. Arrows indicate missense mutations of pathogenic (red) and unknown (blue) significance based on COSMIC[20], as well as nonsense mutations (grey). Note that mutant *KRAS* traces hide under wild-type traces superimposed by wild-type *KRAS* alleles and/or by RNA from tumour or MPE-infiltrating benign somatic cells. Importantly, some mutations were not detected by the analysis software (see letters above mutant trace). Note also multiple mutations in the same patient suggesting a possible multiclonal origin of this MPE. (**c–e**) Patient-derived MPE cell line isolation from eight patients from Institution 1 that were initially tested *KRAS* wild-type. (**c**) Arrow shows focal clonal expansion of cultured MPE cells that gave rise to cell line PB-183. Scale bars, 50 μm. (**d**) PB-183-induced tumour in *NOD/SCID* mouse four weeks after subcutaneous injection of a million cells (*n* = 5). Scale bar, 1 cm. (**e**) Partial Sanger-sequencing traces of *KRAS* cDNA from the initial MPE cells and from two MPE-derived cell lines indicate *KRAS* mutations (red arrows and fonts) that were not identified in the initial samples. Note that even in MPE cell lines mutant *KRAS* traces hide under wild-types traces superimposed by wild-type alleles. Again, the mutation was not detected by the software (see letters above mutant trace).

mutations may be functionally involved in MPE formation (that is, HER2), a postulation that awaits experimental confirmation. To this end, future human studies aimed at identifying genotype–phenotype linkages in various tumours need to be tailored appropriately and need not rely on cross-sectional frequency observation design. Prospectively genotyped, case-matched, and longitudinally observed patient cohorts are more likely to give answers to questions such as the *KRAS*-MPE link proposed here.

In addition to the novel cancer genome-phenotype association, we further show here that mutant *KRAS*-driven MPE is mediated via CCL2-dependent paracrine signalling to CD11b + Gr1 + myeloid cells. The well-studied cell-autonomous effects of mutant *KRAS* conferring addictive proliferation advantages to the tumour cell[35,37] may be complemented by this paracrine axis and may temporally precede its clinical manifestation, since

mutant *KRAS* likely promotes pleural metastasis prior to MPE development[18,19]. As opposed to neutrophil chemoattractants such as CXCL1, CXCL2 (ref. 25), tumour-elaborated CCL2 is a potent monocyte/macrophage mobiliser promoting angiogenesis and metastasis[34,38] and was identified here as the transcriptional target of mutant *KRAS* in tumour cells. This finding complements previous observations implicating CCL2 in mutant *KRAS*-driven inflammation in the lung epithelium[39] and in MPE formation[5,6]. Interestingly, mutant *HRAS* also induces IL-8 signalling[40] and the results imply that different RAS proteins may control distinct chemokine repertoires in order to mobilize defined myeloid cell subsets to tumour sites.

Our present and previous findings[5–7] indicate that pleural tumour-originated CCL2 mobilizes two distinct cell populations from the bone marrow: mast cells and CD11b + Gr1 + cells[41,42], both known to respond to CCL2 (refs 7,25), to facilitate breast

**Table 4 | KRAS mutations in human malignant pleural effusions.**

| | MPEs sequenced (n) | missense KRAS mutations discovered (n) | mutant patients (n) | Known pathogenic mutations | Novel mutations of unknown causality |
|---|---|---|---|---|---|
| **Present study** | 12 | 8 | 5 | G12A, Q61H/E/R | I55M, L56I |

| n(%) | Bulk MPE RNA | MPE cell lines |
|---|---|---|
| KRAS WT | 8 | 3 |
| KRAS MUT | 0 | 5 |

| | n(%) | WT | EGFR MUT | KRAS MUT | P |
|---|---|---|---|---|---|
| **Renaud, S., et al.[19]** | **Bone** | 57(12) | 0(0) | 61(13) | 1.0000 |
| | **Liver** | 39(8) | 8(2) | 9(2) | < 0.0001 |
| | **Brain** | 26(5) | 16(3) | 9(2) | < 0.0001 |
| | **Pleura** | 37(8) | 2(0) | 89(19) | 0.0039 |
| | **Lung** | 89(19) | 1(0) | 22(5) | < 0.0001 |
| | **Adrenal** | 10(2) | 0(0) | 6(1) | < 0.0001 |
| | | 1.0000 | < 0.0001 | < 0.0001 | |

| n(%) | KRAS WT | KRAS MUT |
|---|---|---|
| **Pleura** | 39(8) | 89(19) |
| **Other** | 246(51) | 107(22) |

MPE, malignant pleural effusions; MUT, mutant; WT, wild type.
Present study: top—incidence and type of KRAS mutations detected in 12 human MPE caused by metastatic lung adenocarcinomas from Institution 1. Bottom—summary of KRAS mutations of MPE cell lines isolated from eight patients from Institution 1 that were initially tested KRAS wild-type versus corresponding MPE samples. $P = 0.0256$ by Fischer's exact test.
Renaud et al.[19]: site of recurrence of 481 patients with resected non-small-cell lung cancer according to mutation status[19] shows increased pleural dissemination rates in patients with KRAS-mutations. Top—metastatic site by genotype. $P < 0.0001$ by $\chi^2$-test. P values for comparison with bone metastases or WT tumours by Fischer's exact tests. Bottom: pleural versus any other metastatic site by KRAS genotype. $P < 0.0001$ by Fischer's exact test.
n(%), of patients.

cancer metastasis to the lungs[34], and to sustain tumour growth by promoting angiogenesis[28]. CD11b + Gr1 + cells were previously identified in MPE[42] along with mast cells that were shown to promote MPE by fostering tumour growth and vascular permeability[7]. Here we show that the spleen is an important intermediate organ for MPE development, similar to other tumour models, with its marginal zone functioning as a reservoir for bone-marrow-derived CD11b + Gr1 + cell progenitors that are subsequently rapidly deployed to tumour sites[43,44]. Our experiments, in line with the work of others[26–28,43,44] incriminate the spleen as a pro-tumour organ and suggest that the splenic environment is essential for CD11b + Gr1 + cell recruitment to MPE. As splenectomy provided marked protection to mice against incipient MPE, splenectomy at the time of pleurodesis or catheter placement may yield considerable benefit to patients with MPE, a notion worth exploring.

Finally, we present evidence that mutant KRAS-mediated MPE is actionable by the novel inhibitor of KRAS membrane transport deltarasin, lending hope for clinical targeting of the oncogene in the future[30,35]. Importantly, a CCL2 neutralizing antibody[6,7,33,34] was as effective as deltarasin, strengthening the KRAS-CCL2 connection and indicating that intercepting downstream of mutant oncogene targets may be an alternative to their direct targeting. In addition to the clinical significance of KRAS and other driver mutations of lung and other cancers in MPE that needs to be established, open questions that remain include whether the hypoxic pleural environment impacts MPE development and whether it triggers phenotypic changes in pleural metastasized tumour cells, including the KRAS/CCL2 axis reported here.

In summary, we show that KRAS mutations are causally linked with MPE in mice. We also show that this link rests on a defined innate immune response and that it might be at play in humans with the condition. We believe that this work opens up avenues of potential progress towards aetiologic MPE therapy, by providing preclinical proof-of-concept data on immediate and feasible targeted interventions, such as splenectomy and KRAS and CCL2 blockade, which could provide meaningful benefits to patients with MPE in the future.

## Methods

**Study approval.** Human MPE samples from twenty patients with lung adeno-carcinoma-associated MPE from Institution 1 were obtained and biobanked according to a prospectively placed, standardized and Institutional Ethics Committee-approved protocol (approval number 22699/21.11.2013) that abides by the Declaration of Helsinki. Written informed consent was obtained from each patient. Mouse experiments were carefully designed and were prospectively approved by the Veterinary Administration of the Prefecture of Western Greece (protocol approval numbers 3741/16.11.2010, 60291/3035/19.03.2012, and 118018/578/30.04.2014), and were conducted according to Directive 2010/63/EU (http://eur-lex.europa.eu/LexUriServ/LexUriServ.do?uri=OJ:L:2010:276:0033:0079:EN:PDF).

**Reagents.** Evans' blue and 3-(4,5-dimethylthiazol-2-yl)-2,5-diphenyltetrazolium bromide (MTT) assay powder were from Sigma-Aldrich (St Louis, MO); D-luciferin was from Gold Biotechnology (St Louis, MO); Mouse Gene ST2.0 microarrays and relevant reagents were from Affymetrix (Santa Clara, CA); murine CCL2, CXCL1, and CXCL2 and human CCL2 ELISA kits were from Peprotech EC (London, UK);

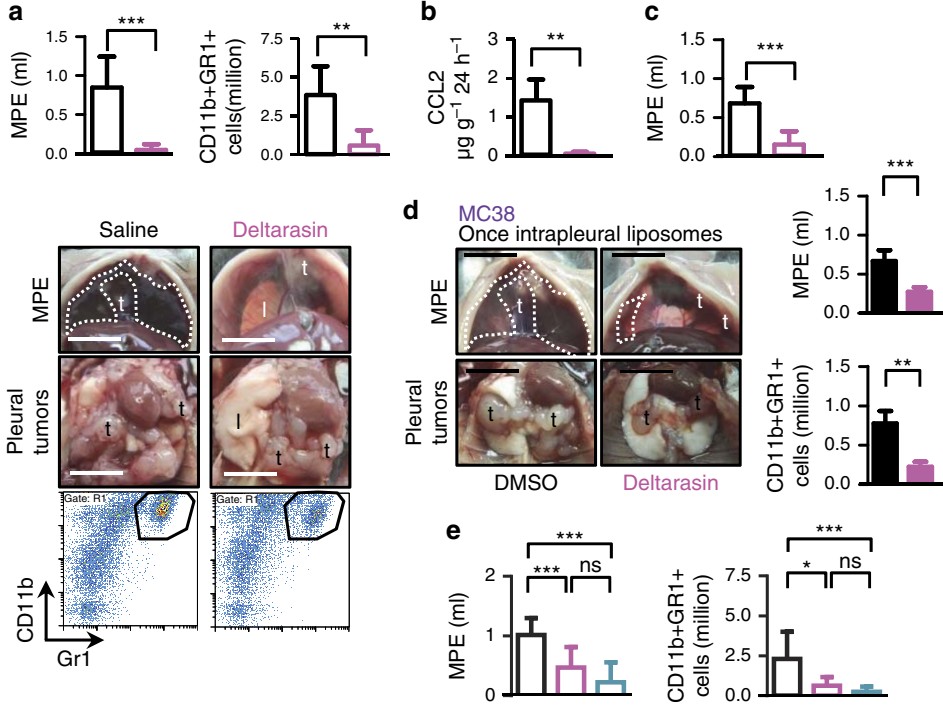

**Figure 5 | Mutant *KRAS*-mediated malignant pleural effusions are actionable. (a)** *C57BL/6* mice received pleural MC38 cells (*ΔKras*G13R), were allowed seven days for pleural tumour development, and were randomized to daily intraperitoneal saline (100 μl) or deltarasin (15 mg kg$^{-1}$) treatments. Shown are data summaries of malignant pleural effusion (MPE) volume and CD11b + r1 + cells (both $n = 8$/group), representative images of pleural effusions (dashed lines) and tumours (t), and representative dotplots of CD11b + Gr1 + cells (polygon gates) at day 13 post-MC38 cells. Scale bars, 1 cm. **(b)** MC38 cells were treated *in vitro* with saline or deltarasin (15 μgml$^{-1}$). Shown is CCL2 secreted at 24 h ($n = 5$/group). **(c)** *NOD/SCID* mice received pleural LTP A549 cells (*ΔKRAS*G12S), were allowed 14 days for pleural tumour development, and were randomized to daily intraperitoneal saline (100 μl) or deltarasin (15 mg kg$^{-1}$) treatments. Shown is data summary of MPE volume at day 30 post-tumour cells. **(d)** *C57BL/6* mice received pleural MC38 cells followed by a single intrapleural injection of liposomes containing 1% DMSO or 15 mg kg$^{-1}$ deltarasin in 1% DMSO at day 7 post-tumour cells. Shown are representative images of pleural effusions (dashed lines) and tumours (t), and data summaries of MPE volume ($n = 15$–16/group) and CD11b + Gr1 + cells ($n = 9$/group) at day 13 post-MC38 cells. Scale bars, 1 cm. **(e)** *C57BL/6* mice received pleural PANO2 cells stably expressing mutant *Kras* vectors (p*ΔKras2A* or p*ΔKras2B*), were allowed 4 or 14 days, respectively, for pleural tumour development and were then randomized to intraperitoneal treatment with daily saline plus IgG2a antibody every three days (50 mg kg$^{-1}$ in 100 μl saline), daily deltarasin (15 mg kg$^{-1}$ in 100 μl saline), or anti-CCL2 antibody every three days (50 mg kg$^{-1}$ in 100 μl saline). Shown are data summaries of MPE volume ($n$ 27, 10, and 20 mice/group, respectively) and CD11b + Gr1 + cells ($n = 24$, 8, and 14/group, respectively) at day 14 post-tumour cells. Data are presented as mean ± s.d. ns, *, **, and ***: $P > 0.05$, $P < 0.05$, $P < 0.01$, and $P < 0.001$ for the indicated comparisons by Student's t-test (**a-d**) or one-way ANOVA with Bonferroni post-tests (**e**).

**Table 5 | Incidence of malignant pleural effusions in *KRAS*-targeted mice.**

| | Treatments | No MPE | MPE | P value |
|---|---|---|---|---|
| Experiment from Fig. 5a MC38-induced MPE daily intraperitoneal treatments installed at day 7 post-MC38 cells | Saline (100 μl) | 1 | 7 | — |
| | Deltarasin (15 mg kg$^{-1}$ in 100 μl saline) | 7 | 1 | 0.0101 |
| Experiment from Fig. 5c LTP A549-induced MPE daily intraperitoneal treatments installed at day 14 post-MC38 cells | Saline (100 μl) | 0 | 10 | — |
| | Deltarasin (15 mg kg$^{-1}$ in 100 μl saline) | 5 | 4 | 0.0108 |
| Experiment from Fig. 5d MC38-induced MPE once intrapleural treatment at day 7 post-MC38 cells 100 μl injectate volume | Liposomes (saline 1% DMSO) | 2 | 14 | |
| | Liposomes (15 mg kg$^{-1}$ deltarasin in saline 1% DMSO | 8 | 7 | 0.0151 |
| Experiment from Fig. 5e PANO2 p*ΔKras2A/2B*-induced MPE intraperitoneal treatments installed at day 4 or 14 days post-tumour cells, respectively | Daily saline (100 μl) + IgG2a every three days (50 mg kg$^{-1}$ in 100 μl saline) | 0 | 27 | — |
| | Daily deltarasin (15 mg kg$^{-1}$ in 100 μl saline) | 3 | 7 | 0.0157 |
| | α-CCL2 every 3 days (50 mg kg$^{-1}$ in 100 μl saline) | 14 | 6 | < 0.0001 |

CCL, C–C motif chemokine ligand; DMSO, dimethyl sulfoxide; LTP A549, long-term passaged A549 cells having lost the Y chromosome; MC38, *C57BL/6* colon adenocarcinoma; MPE, malignant pleural effusion; PANO2, *C57BL/6* pancreatic adenocarcinoma; *Δ*, mutant.
MPE incidence of *C57BL/6* mice that received *KRAS*-mutant pleural tumour cells followed by deltarasin or anti (α)-CCL2 treatments. Shown are numbers of mice ($n$) and probability ($P$) values for comparison with controls by Fischer's exact test. $P < 0.0001$ for overall comparison of experiment from Fig. 5e by $\chi^2$-tests.

primer sets and antibodies are listed in Supplementary Tables 1 and 2, respectively; RAS activation assay was from Merck Millipore (Darmstadt, Germany); deltarasin was from MedChem Express (Princeton, NJ) and from Cayman Europe (Tallinn, Estonia); anti-mouse CCL2 neutralizing antibody, as well as IgG2a control antibody were from Oncology Discovery Research, Janssen R&D LLC (Radnor, PA)[33,34]; 1, 2-Distearoyl-sn-glycero-3-phosphocholine, phosphatidylglycerol and cholesterol were from Avanti Polar Lipids, Inc. (Alabaster, AL).

**Mice.** C57BL/6 (#000664), NOD/SCID (#001303), CAG.Luc.eGFP (#008450), and Ccr2$^{-/-}$ (#004999) mice from Jackson Laboratories (Bar Harbor, ME) were bred in the University of Patras Center for Animal Models of Disease. All experiments entailing murine cell lines were done using mice on the C57BL/6 background or CAG.Luc.eGFP mice backcrossed >F12 to the C57BL/6 background. All experiments entailing human cell lines were done using mice on the NOD/ShiLtJ background. Nine hundred and seventy-five sex-, weight (20–25 g)- and age (6–12 week)-matched male and female (50% of mice from each sex were enrolled in each experimental arm) experimental mice were used for these studies. The exact animal numbers per experiment are given in Tables 1, 2 and 5 and in the Legends to Figures.

**Cells.** C57BL/6 mouse B16F10 skin melanoma and PANO2 pancreatic and Lewis lung carcinomas (LLC), as well as human SKMEL2 skin melanoma, A549 lung and HT-29 colon adenocarcinomas were from the National Cancer Institute Tumour Repository (Frederick, MD); human HEK293T embryonic kidney cells were from the American Type Culture Collection (Manassas, VA); C57BL/6 mouse MC38 colon adenocarcinoma cells were a gift from Dr Barbara Fingleton (Vanderbilt University, Nashville, TN, USA)[6,7,45], C57BL/6 mouse AE17 malignant pleural mesothelioma cells from Dr Y.C. Gary Lee (University of Western Australia, Perth, Australia)[46], and human LTP A549 cells that have suffered chromosome Y loss from Dr Haralabos P. Kalofonos (University of Patras, Greece). Primary lung adenocarcinoma cells from C57BL/6 and FVB mice (CULA and FULA cells, respectively) were generated as described elsewhere[23]. Briefly, C57BL/6 and FVB mice received ten and four consecutive weekly intraperitoneal injections of urethane (1 g kg$^{-1}$) and were killed ten months later. Lung tumours were isolated under sterile conditions, strained to single cell suspensions, and cultured for >100 passages over two years. Primary airway cells were derived by culturing stripped murine tracheal epithelium. Cell lines were authenticated annually using the short tandem repeat method, microarray, and Sanger sequencing and were tested for Mycoplasma Spp. biannually by PCR using designated primers (Supplementary Table 1). All cell lines were cultured at 37 °C in 5% CO$_2$-95% air using full culture medium (DMEM supplemented with 10% FBS, 2 mM L-glutamine, 1 mM pyruvate, 100 U ml$^{-1}$ penicillin, and 100 mg ml$^{-1}$ streptomycin). For in vivo injections, cells were collected with trypsin, incubated with Trypan blue, counted by microscopy in a haemocytometer, their concentration was adjusted in PBS, and cell were injected through a left intercostal space or in the skin, as described elsewhere[5–7]. Only 95% viable cells were used for in vivo injections.

**Sequencing plasmids and microarrays.** Total cellular RNA was isolated using Trizol (Invitrogen, Thermo Fisher Scientific, Waltham, MA) followed by RNAeasy column purification and genomic DNA removal (Qiagen, Hilden, Germany). One µg purified total RNA was reverse transcribed using an Oligo(dT)$_{18}$ primer and Superscript III (Invitrogen, Thermo Fisher Scientific, Waltham, MA) according to the manufacturer's instructions. For sequencing reactions, Kras, Egfr, Braf and Pik3ca cDNAs (or parts of these cDNAs) were amplified in PCR reactions using the corresponding primers (Supplementary Table 1) and Phusion Hot Start Flex polymerase (New England Biolabs, Ipswich, MA). cDNA fragments were purified with NucleoSpin gel and PCR clean-up columns (Macherey-Nagel, Düren, Germany) and were directly Sanger-sequenced with their corresponding forward and reverse primers by VBC Biotech (Vienna, Austria). For RNA interference, the following proprietary lentiviral shRNA pools of three were obtained from Santa Cruz Biotechnology (Palo Alto, CA): random control shRNA (shC, sc-108080-V), GFP control (sc-108084-V), and anti-Kras.shRNA (shKras, sc-33876-V). Anti-Kras lentiviral shRNA target sequences were: 5′-CTACAGGAAACAAGTAGTA-3′, 5′-GAACAGTAGACACGAAACA-3′ and 5′-CCATTCAGTTTCCATGTTA-3′. For this study, the following new plasmids were constructed in-house and were deposited with Addgene (https://www.addgene.org/Georgios_Stathopoulos/), accompanied by their full sequence files: (i) a pMIGR1-based puromycin resistance bicistronic retroviral expression vector (Addgene ID 64335) was constructed by replacing the eGFP sequences of pMIGR1 vector downstream of IRES with puromycin resistance sequences; (ii) a pMIGR1-based hygromycin resistance bicistronic retroviral expression vector (Addgene ID 64374) was constructed by replacing the eGFP sequences of pMIGR1 vector downstream of IRES with hygromycin resistance sequences. Both puromycin and hygromycin resistance genes were subcloned by restriction enzymes from available construct; (iii) the derivatives of these vectors, namely eGFP.retro.puro (Addgene ID 64336), eGFP.retro.hygro (Addgene ID 64375), eGFP.KRAS$^{wt}$-2B retro.puro (Addgene ID 64371), eGFP.KRAS$^{G12C}$-2B.retro.puro (Addgene ID 64372), eGFP.KRAS$^{G12C}$-2B.retro.hygro (Addgene ID 64376), eGFP.KRAS$^{G12C}$-2A.retro.puro (Addgene ID

64373). Murine Kras$^{G12C}$ isoform 2A and 2B cDNAs were amplified using a first stand synthesis reverse transcription reaction from 1 µg total LLC RNA using specific primers (Supplementary Table 1) and Phusion Hot Start Flex polymerase (New England Biolabs, Ipswich, MA). Similarly, wild type Kras2B cDNA was amplified using a first stand synthesis reverse transcription reaction from 1 µg total PANO2 cellular RNA.The corresponding cDNAs were cloned into a peGFP.C1 vector (Clontech, Mountain View, CA) between BglII and EcoRI restriction sites, in-frame with eGFP and were verified by sequencing. The eGFP, as well as the wild type eGFP.KRAS2B and mutant eGFP.KRAS$^{G12C}$2A and 2B-encoding sequences were restricted out from the corresponding peGFP.C1-based constructs described above using enzymes AgeI and SmaI, were gel purified, and were subcloned into the (AgeI/HpaI-restricted) retroviral bicistronic expression vectors upstream of IRES-puromycin (ID 64336) and/or IRES-hygromycin (ID 64375) resistance genes, respectively. Retroviral particles were obtained by transfecting a 3 mm well of confluent HEK293T cells with the desired bicistronic retroviral vectors (expressing either eGFP or wild type or mutant eGFP.KRAS) along with VSV-G envelope expressing plasmid pMD2.G (Addgene ID12259) and pCMV-Gag-Pol expressing the retroviral structure proteins (Cell Biolabs Inc, San Diego, CA) at 1.5:1:1 mass stoichiometry via the CaCl$_2$/BES method. After two days, the cultured medium (2 ml for each retroviral type) was collected and passed through a 45 µM filter to remove cellular debris. This supernatant material was supplemented with 8 ml medium and was subsequently overlaid on a 100 mm plate containing 70% confluent mouse cancer cells. After 48 h, the medium was removed and the cells were incubated with selection medium (full cell culture medium supplemented with either 2–10 µg ml$^{-1}$ puromycin or 50–100 µg ml$^{-1}$ hygromycin). Stable clones were selected and subcultured. All other cell lines were transfected with standard DNA amounts using X-Fect (Clontech, Mountain View, CA).

**Mouse experiments.** Experimental pleural carcinomatosis was induced by pleural delivery of 1.5 × 10$^5$ murine cancer cells, 10$^6$ human cancer cells or 3 × 10$^6$ HEK293T cells. MPE models and splenectomy have been described elsewhere[5–7]. For bioluminescence imaging, mice were serially imaged on a Xenogen Lumina II and data were analysed using Living Image v.4.2 (Perkin-Elmer, Waltham, MA), after delivery of 1 mg intravenous D-luciferin (Gold Biotechnology, St Louis, MO) by retro-orbital injection. For splenocyte give-back, spleens were removed under sterile conditions from CAG.Luc.eGFP donors (n = 3 per group), 13 days after intrapleural injection with saline or tumour cells. Single-cell suspensions were prepared by passing spleens through 70 µm nylon cell strainers (BD Biosciences, Bedford, MA), followed by delivery of 100 µl saline containing 5 × 10$^6$ splenocytes to splenectomized hosts.

**Mouse models.** For induction of malignant pleural carcinomatosis with or without a MPE, mice were anaesthetized using isoflurane inhalation and received intrapleural injections of 100 µl PBS containing 150,000 murine, 1,000,000 human cancer cells or 3,000,000 HEK293T cells. Mice were killed when moribund for survival and pleural fluid and tumour analyses. Mice with pleural fluid volume ≥ 100 µl (equals the initial injection volume) were judged to have a MPE ('wet' pleural metastasis, that is, active pleural fluid exudation) and were subjected to pleural fluid aspiration, whereas animals with pleural fluid volume < 100 µl were judged not to have a MPE ('dry' pleural carcinomatosis) and were subjected to pleural lavage. For this, 1 ml normal saline was injected intrapleurally and was withdrawn after 30 s. Following pleural fluid or lavage retrieval, the chest was opened and pleural tumours were stripped and weighed. For subcutaneous tumour formation, mice received 1,000,000 subcutaneous cancer cells and were followed longitudinally. Three vertical tumour dimensions ($\delta$1, $\delta$2 and $\delta$3) were monitored serially and tumour volume was calculated using the formula $\pi \times \delta1 \times \delta2 \times \delta3/6$. Both models have been described elsewhere[5–7].

**Cytology histology and microscopy.** Before further processing, MPE fluids were subjected to red blood cell lysis in ten-fold excess red blood cells lysis buffer (155 mM NH$_4$Cl, 12 mM NaHCO$_3$, 0,1 mM EDTA). Total pleural cell counts were determined microscopically on a haemocytometer and cytocentrifugal specimens (5 × 10$^4$ cells each) of total pleural fluid cells (or of CD45 + CD11b + Gr1 + sorted cells) were fixed with methanol for 2 min. Cells were stained with May–Grünwald working solution (May–Grünwald stain in 1 mM Na$_2$HPO$_4$, 2.5 mM KH$_2$PO$_4$, pH = 6.4) for 6 min, then with Giemsa working solution (Giemsa stain in 2 mM Na$_2$HPO$_4$, 5 mM KH$_2$PO$_4$, pH = 6.4) for 40 min, then washed with H$_2$O, and dried. Slides were mounted with Entellan (Merck Millipore, Darmstadt, Germany) and coverslipped and were microscopically analysed. For flow cytometry and fluorescence-activated cell sorting (FACS) staining, 10$^6$ nucleated pleural fluid cells suspended in 50 µl FACS buffer (PBS supplemented with 2% FBS and 0,1% NaN$_3$) were stained with the indicated antibodies according to manufacturer's instructions (Supplementary Table 2) for 20 min in the dark, were washed with FACS buffer from excess antibody, and were resuspended in 1 ml FACS buffer for further analysis. Spleens fixed in 10% formaldehyde overnight were embedded in paraffin and stored at room temperature. Five-micrometre paraffin sections were mounted on glass slides. Pleural tumours fixed in 10% paraffin or in 4% paraformaldehyde overnight were embedded in paraffin or in optimal cutting temperature (OCT; Sakura, Tokyo, Japan) and were stored at room temperature or − 80 °C,

respectively, till further analyses. Five-micrometre-thick paraffin or 10-μm-thick cryosections were mounted on glass slides. Sections were labelled using the indicated antibodies (Supplementary Table 2) and were counterstained with Hoechst 33258 or with Envision colour development system (Dako, Carpinteria, CA). Immunoreactivity was quantified as described previously[5–7]. Bright-field and fluorescent microscopy were carried out using either an AxioObserver D1 inverted microscope (Zeiss, Jena, Germany) or an SP5 confocal microscope (Leica, Heidelberg, Germany).

**Flow cytometry and cell sorting.** A CyFlow ML instrument with FloMax Software (Partec, Münster, Germany) was used for cell cytometry, sorting, and data analysis. CD45 + CD11b + Gr1 + cells were identified, enumerated by true volumetric counting, and their total numbers were calculated as fraction of total MPE cell counts, or were sorted and collected in FACS buffer at $5 \times 10^4$ cells per ml.

**Cellular assays.** *In vitro* cancer cell proliferation was determined using the 3-(4,5-dimethylthiazol-2-yl)-2,5-diphenyltetrazolium bromide (MTT) assay. For this, $2 \times 10^4$ cells per well were plated onto 96-well plates. Daily thereafter, 15 μl of 5 mM MTT working solution in PBS was added to wells to be measured that day. The plate was incubated for 4 h at 37 °C in a 5% $CO_2$ humidified incubator followed by addition of 100 μl acidified isopropanol per well for sediment solubilization and absorbance measurement at 492 nm on an MR-96 A photometer (Mindray, Shenzhen, China). For soft agar colony formation assay, $7.5 \times 10^3$ cells were plated in 60 mm culture vessels in semi-solid 0.7% agarose in full culture medium and were incubated for 30 days at 37 °C in a 5% $CO_2$ humidified incubator. 2 ml fresh culture medium was added to each vessel biweekly. After incubation, 500 μl MTT working solution was added to each vessel and plates were dried, inverted, photographed and colonies were counted.

**Vascular permeability assays.** Mice with MPE received 0.8 mg intravenous Evans' blue and were killed after 1 h for determination of MPE levels of the albumin-binding dye[5–7]. Intradermal injections of 50 μl cell-free pleural lavage or MPE were performed at different spots of the shaved dorsal mouse skin followed immediately by Evans' blue injections as above, and euthanasia, skin inversion and imaging after one hour. The surface area of dye leak was determined using Fiji academic freeware (http://fiji.sc/Fiji), as described elsewhere[5–7].

**Chick chorioallantoic membrane assay.** Fertilized White Leghorn chicken eggs, as soon as embryogenesis started, were placed for incubation under constant humidity at 37 °C. On day 4, a square window was opened in the shell and then sealed with adhesive tape. On day 9, a 1 cm² rubber O-ring was placed on the surface of the CAM and 50 μl PBS or cell-free MPE or pleural lavage supernatants were added inside the ring's restricted area. After 48 h, CAM tissues were fixed in Carson's solution (saline-buffered formalin) and angiogenesis was evaluated using Fiji.

**Real-time quantitative PCR and microarray hybridization and analysis.** qPCR was performed using first strand synthesis reactions along with the indicated primers and KAPA SYBR FAST qPCR Kit (Kapa Biosystems, Wilmington, MA) in a StepOne cycler (Applied Biosystems, Carlsbad, CA). The CT values from triplicate qPCR reactions were extracted from the StepOne cycler (Applied Biosystems, Carlsbad, CA) onto Excel spreadsheets and were analysed with the relative quantification method $2^{-\Delta\Delta CT}$. The expression level of a given mRNA per sample/condition was determined relatively to reference gene mRNA levels. For microarray, triplicate cultures of $10^6$ cells (for each cell line/condition) were subjected to RNA extraction as above. Five μg of pooled total RNA were tested for RNA quality on an ABI2000 bioanalyser (Agilent Technologies, Sta. Clara, CA), labelled, and hybridized to GeneChip Mouse Gene 1.0 or 2.0 ST arrays according to manufacturer's instructions (Affymetrix, Sta. Clara, CA). For microarray analysis, the Affymetrix Expression Console was used (parameters: annotation confidence, full; summarization method: iter-PLIER include DABG; background: PM-GCGB; normalization method: none), followed by normalization of all arrays together using a Lowess multi-array algorithm. Intensity-dependent estimation of noise was used for statistical analysis of differential expression. Unsupervised hierarchical clustering of microarray data and Venn diagrams were performed using the MatLab 2014 program. Microarray data are publicly available at the GEO database (http://www.ncbi.nlm.nih.gov/geo/; Accession ID: GSE58190 and GSE85021).

**RAS activation assay and immunoblotting.** Total protein extracts were prepared by lysing $10^7$ cultured cells in $Mg^{2+}$ Lysis/Wash buffer (25 mM HEPES pH = 7.5, 150 mM NaCl, 1% NP-40, 10 mM $MgCl_2$, 1 mM EDTA, 2% glycerol). RAS activation assay was performed with RAS Activation Assay Kit (Merck Millipore, Darmstadt, Germany) by incubating total cell lysates with Raf-1 Ras-Binding Domain (RBD)-coated agarose beads according to the manufacturer's instructions. Total protein extracts and Raf-1 RBD agarose bead samples were analysed using 6–20% SDS–PAGE followed by standard blotting to PVDF membranes (Merck Millipore, Darmstadt, Germany). Membranes were incubated with primary

antibodies, followed by incubation with the appropriate HRP-conjugated secondary antibodies at the manufacturers' indicated dilutions (Supplementary Table 2), and were visualized by chemiluminescence film exposure, using the enhanced chemiluminescence substrate (Merck Millipore, Darmstadt, Germany). Full uncropped immunoblots are shown in Supplementary Figures 8–10.

**Enzyme-linked immunosorbent assays (ELISA).** CCL2, CXCL1 and CXCL2 levels of cell culture supernatants, as well as CCL2 levels of cell-free MPE and corresponding sera were determined using dedicated murine and human ELISA kits according to the manufacturer's instructions (Peprotech, London, UK and R&D, Minneapolis, MN).

**Bone marrow transplantation.** For adoptive bone marrow replacement, *C57BL/6* hosts received ten million intravenous bone marrow cells flushed from the femurs and tibias of *CAG.Luc.eGFP* donors, 12 h after total-body irradiation (1,100 rad), as described elsewhere[6,7]. One mouse in each experiment was not engrafted (sentinel) and was observed till moribund between days 5 and 15 post-irradiation. The mice were left to recover at least for one month before subjecting them to experimentally induced MPE.

**Liposome preparation and physicochemical characterization.** Deltarasin-encapsulating liposomes were prepared by the DRV technique as described elsewhere[31,32] by freeze drying 30 mg of empty DSPC/PG/Chol (9:1:5 (mol:mol:mol)) unilamelar sonicated vesicles with 1 ml of deltarasin solution (5 mg ml⁻¹) in PBS, or plain PBS (for empty liposomes), followed by controlled re-hydration. The liposome size was decreased by extrusion though Lipo-so-fast extruder, polycarbonate membranes (Avestin Europe) with 400 nm diameter pore. Liposomes lipid concentration, size distribution and surface charge (zeta-sizer, Malvern) were estimated as reported elsewhere[31,32], as well as drug encapsulation efficiency after measuring the non-liposomal drug absorption at 284 nm.

***In vivo* drug treatments.** Treatments were initiated 4–14 days post-mouse and 14 days post-human pleural tumour cell injections. Deltarasin (15 mg kg⁻¹ in 100 μl PBS), or 100 μl PBS were given daily intraperitoneally. Anti-mouse CCL2 or IgG2a control antibodies were delivered intraperitoneally at 50 mg kg⁻¹ in 100 μl PBS every three days[6,7]. Deltarasin-encapsulating liposomes were prepared as described above and elsewhere[31,32] and were delivered intrapleurally seven days post-intrapleural MC38 cells.

**Human MPE.** MPE cell concentrates from patients with lung adenocarcinoma-associated MPE from Institution 1 ($n = 20$) were obtained and biobanked according to standard protocols, were handled similar to mouse MPE samples, and were subjected to RNA extraction, cDNA synthesis, PCR with human *KRAS*-specific primers (Supplementary Table 1), and direct Sanger sequencing. Some MPE cells were cultured for one month, tumor cell clones were picked and pooled, were inoculated into the flank of *NOD/SCID* mice to test their tumorigenicity, and were sequenced as above.

**Statistics.** Sample size was calculated using G*power (http://www.gpower.hhu.de/;[47]) assuming errors $\alpha = 0.05$ and $\beta = 0.05$, and effect size $d = 1.5$. Experiments were done in a fractionated fashion until statistical significance ($P < 0.05$ with $\alpha < 0.05$) was reached or ruled out ($P > 0.05$ with $\beta < 0.05$). No data were excluded from analyses and controls and intervention animals were enrolled as necessary per interim power analyses. Greater numbers of animals were added to follow-up experiments in groups where outliers increased the dispersion of the data, generating uneven experimental groups. Animals were allocated to treatments by alternation, and transgenic animals were enrolled case-control-wise. Data acquisition was blinded on samples previously coded by a non-blinded investigator. All data were examined for normality by Kolmogorov–Smirnov test and were found to be normally distributed. Values are given as frequencies, mean ± s.e.m., or mean ± s.d., as indicated. Sample size ($n$) refers to biological replicates, except from cell line qPCR data, where $n$ refers to technical replicates. Differences in frequencies were examined by $\chi^2$ or Fischer's exact tests, as appropriate. Differences in means were examined by two-tailed Student's *t*-test, or one-way or two-way ANOVA with Bonferroni post-tests, as appropriate. Survival proportions were examined by Kaplan–Meier analysis using the log-rank test for overall and pairwise comparisons. $P$ values are two-tailed, and $P < 0.05$ was considered significant. Analyses and plots were done on Prism v5.0 (GraphPad Software, La Jolla, CA).

**Data availability.** The microarray data have been deposited at the Gene Expression Omnibus database (http://www.ncbi.nlm.nih.gov/geo/) under the accession codes GSE58190 and GSE85021. The vectors have been deposited at the Addgene plasmid depository (http://www.addgene.org/Georgios_Stathopoulos/) and plasmid ID's are given in the text. Primary mouse lung adenocarcinoma cells were deposited at the Laboratory for Molecular Respiratory Carcinogenesis (LMRC) core cell line facility (http://www.lmrc.upatras.gr/) and are available upon request (lmrc@upatras.gr). The authors declare that all the other data supporting the

findings of this study are available within the article and its Supplementary Information files and from the corresponding author upon reasonable request.

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

## Acknowledgements

This work was supported by European Research Council 2010 Starting Independent Investigator and 2015 Proof of Concept Grants (#260524 and #679345, respectively; to G.T.S.). We thank the University of Patras Centre for Animal Models of Disease and Advanced Light Microscopy Core for experimental support. The authors also thank Drs Barbara Fingleton (Vanderbilt University, Nashville, TN, U.S.A.), Y.C. Gary Lee (University of Western Australia, Perth, Australia), and Haralabos P. Kalofonos (University of Patras, Greece) for the kind provision of cell lines critical to this work.

## Author contributions

T.A., A.D.G. and A.C.K. conceived, designed and carried out most experiments, analysed the data, provided critical intellectual input and wrote the paper draft; N.I.K., D.K., M.V. and M.P. analysed the microarray data, performed PCR, qPCR and Sanger sequencing and analysed the relevant data; M.S., I.L. the D.E.Z. performed immunohistochemistry and microscopy; performed CCR2 and CD68 immunolocalization; E.N. carried out KRAS silencing; M.V. and N.S. carried out in vivo experiments; A.P. and D.K. performed mutant KRAS overexpression; K.P. and S.G.A. prepared liposomes; A.V., A.C.K., A.M., N.I.K., I.P., L.K., M.V. and G.T.S. established clinical protocols and obtained and processed clinical samples; V.H. performed microarrays; P.S. carried out flow cytometry; S.M., O.E., L.A.S. and I.P. provided intellectual input and analytical tools related to KRAS/CCL2 signalling; D.K. performed total body irradiation; A.M. and G.T.S. conceived the idea, supervised the study, designed and carried out experiments, analysed data, wrote the paper and guarantee the study's integrity. All authors concur with the submitted manuscript.

## Additional information

**Competing interests:** L.A.S. is an employee of the company that produces the anti-CCL2 antibodies. The remaining authors declare no competing financial interests.

**Publisher's note**: 

