## [Peer Review File · Nature Communications]

Reviewers' comments:

Reviewer #1 (Remarks to the Author):

A. In this manuscript the authors consider the relationship between tumor expression of KRAS and the development of malignant pleural effusion (MPE). They also examine the role of CCL2, CD11b+/Gr1+ cells and mast cells in this process. They observe that KRAS positive tumors are more likely to induce MPE, that reduced KRAS expression is associated with reduced MPE and that MPE occurs by a mechanism associated with tumor cell expression of CCL2, mast cells and the recruitment of CD11b+/Gr1+ positive cells. Notably, pharmacological KRAS inhibition reduces MPE.

B. MPE is a serious clinical problem that requires attention. The role of KRAS has not previously been examined in this setting. However the role of mast cells and the predominance of CD11b+/Gr1+ cells in the infiltrate has previously been observed, as referenced by the authors.

C. The tumour models chosen for study were an assortment of different cell types from a variety of tissue sources, several of those that induced MPE were of an airways epithelial origin (eg LLC and A549) a more systematic examination of specific types of tumor cells would have been more convincing. There are many databases of tumour profiles and clinical information which could have been used to determine if the association between KRAS and MPE held true on a larger clinical scale.

The mast cell dependency of neutrophil and in some cases monocyte recruitment has been demonstrated in many situations. However, the use of a single type of mast cell deficient mouse together with a reconstitution protocol which, uses a very high dose of mast cells and does not allow the pleural mast cells to differentiate into the type of mast cell found locally in this setting before responding to tumours, limits the interpretation of the data provided.

CD11b+/Gr1+ cells will include a number of cell types, likely mostly neutrophils in this setting, only a proportion of which will have MDSC characteristics. The interpretation that these cells are mainly MDSC is not justified. Further markers and/or functional data would be required. The quality of the single micrograph showing MPE associated infiltrating cells is also very poor. A better analysis of the proportions of neutrophils, monocytes and eosinophils in the MPE would be informative.

Given the impact of KRAS on metabolic activities of tumour cells, it may be important to distinguish between the ability of KRAS to allow tumours to thrive in a low oxygen environment where they promote MPE from any CCL2 production associated impact of KRAS on MPE induction. Use of CCR2 deficient animals, as hosts, could help address this question.

D. More details should be provided on the statistical approach used in each figure. In some cases, it is unclear if experiments were replicated or if the results presented are from a single experiment using several mice. In other situations the statistical interpretation does not seem to match the data as presented. For example in the final panel of figure 6D (1st bar vs 3rd bar and 2nd bar vs 3rd bar).

E. With the data provided, while the conclusion can be drawn that KRAS expression enhances MPE in some animal tumour models it is unclear how this is occurring and whether this is a feature of human disease.

F. Use of CCR2^{-/-} mice. Use of systematic approach to examine association between KRAS expression and MPE. Clarification of data replication and analysis. Improved definition of CD11b+/Gr1+ cell types (see above comments)

G. References are appropriate

H. Some over interpretation of data from murine models. Especially with respect to role of mast cells CCL2 and MDSC. In order to suggest this is a feature of clinical disease, need much more data from a systematic study of human tumors. The discussion is overly long and complex. The figure legends do not give sufficient information

Reviewer #2 (Remarks to the Author):

Dear Authors,

This is an elegant series of well conducted experiments reviewing the mechanism behind MPE formation in a mouse model of MPE, including the use of human tumour cell lines. The experiments appear to show that KRAS mutation is both necessary and sufficient for MPE formation, and that this acts through a MC dependant manner, requiring splenic trafficking. MPE formation appears blocked by a deltarasin.

This is a highly novel area of substantial clinical interest. The methods used appear sound and the data is largely clearly presented (I think too many figures are presented and the main figures only should be presented in the main manuscript, maximum of 5).

Conclusions made by the authors appear largely valid, however, I have several areas of comment:

1. As above, there are too many figures and the authors should choose the most relevant 5 figures and present these only with the others moved to online.
2. A key issue (which the authors address) is that currently in human treatment, KRAS mutation is not commonly found. Some rather weak arguments are presented as to why this should be, but the authors experiments appear to show that KRAS mutation is absolutely required for MPE formation - the other potential conclusion (in addition to the authors which is that our clinical assays for KRAS are not yet sensitive enough) is that this pathway is only relevant in the presented mouse model, and this limitations should be discussed. If KRAS was so essential in human MPE, we would expect to see different KRAS results from biopsy.
3. While patients with and without pleural effusion from malignancy exist, the vast majority (I would estimate 95%) are associated with effusion. This again suggests that some form of KRAS mutation should be detected in nearly all MPE / malignant pleural patients and the authors should discuss this.
4. MPE is a clinically heterogenous set of diseases from a number of primary sites - this may mean that some / all of the pathway delineated is relevant to some patients and not others.
5. The last figure concludes beyond the data and is speculative - please remove completely.
6. I would encourage the authors to conclude within the data and suggest further steps necessary in human MPE to prove their theories.

Reviewer #3 (Remarks to the Author):

The authors found KRAS-dependent CCL2/CCR2 paracrine signaling, resulting in recruitment of

bone marrow derived MC and subsequently of splenic residing CD11b+Gr1+ MDSC into the tumor afflicted pleural space and accumulation of pleural effusion.

These are new and interesting findings from a series of cell lines and in vivo animal studies.

1. The animal numbers of each group are quite variable, such as n=8-27/group in Figure 7. This is not the standard of experimental design. Could the authors describe what animal numbers of each group in the design are?

2. MDSC play an important role in immunosuppression in tumor microenvironment. What is the mechanism that CD11b+Gr1+ MDSC recruited into pleural space causes inflammation and angiogenesis shown in the Supplementary Figure 1?

3. The authors concluded that mEGFR is neither sufficient nor required for experimental MPE induction by the results of Figures S4A and S4B. This is quite contradictory to clinical observation that MPEs occur in every molecular subtypes (K-Ras mutation, EGFR mutation, ALK translocation, ROS-1 translocation, etc).. and histological subtypes (small cell lung cancer, adenocarcinoma, squamous cell carcinoma) of lung cancers. The authors comment in the discussion that mutant KRAS bearing tumors cells causing MPE would escape detection in focal pleural tumor tissue biopsies and low-volume pleural fluid aspirates. This is quite difficult to understand in the genomic era that sensitive methods (esp NGS) been used to find the driver genes for various targeted therapies, which are listed in guidelines of many organizations. In addition, the authors used mEGFR overexpression in B16F10 cell to show no MPE formation. The EGFR protein expression of the B16F10pΔEGFR was quite low in supplementary figure 2g. The experiment that knockdown of EGFR in mutant KRAS cell does not affect the activity of KRAS nor decrease the amount MPE can't be the evidence that mutant EGFR does not form MPE.

4. The supplementary figure S3c does not support the statement -"manipulation of KRAS signaling did not result in obvious change in tumor cell proliferation" in the Results.

Reviewer #4 (Remarks to the Author):

The authors use a large series of mouse based experiments to show essential roles for mutant K-ras, CCL2, CD11b+/Gr1+ cells, and mast cells in the development of malignant pleural effusion (MPE). There is an impressive number of cell lines employed. The data is of high quality and appears quite reproducible. A series of experiments using splenectomy models, bone marrow tagged cells, Ccr2-/- and mast cell-deficient mice are provided to uncover the operative mechanisms at play. In summary, this study includes a large amount of data to support the main conclusions and is quite novel. However, there are some deficiencies in the study that must be addressed.

1. Lack of human correlation. The authors attempt to explain that this wasn't done because it would be overly difficult. No human study is straight forward, but this would be far from difficult to conduct. At most centers in the US, all stage IV lung cancer patients undergo genetic testing for EGFR and ALK. The remaining DNA could be used to genotype KRAS. Since these are all stage IV patients, many of them will have MPE. It should not be difficult to establish MPE rates for KRAS mutant vs. EGFR mutant vs WT for both. The way the current study is written suggests that KRAS is required for MPE. It is difficult to imagine that all MPE is related to KRAS in human disease.

2. Inclusion of an EGFR mutant cell line. The authors do a great job of including numerous cell lines of both mouse and human origin. One concern is that they are of many different histologies. It is important to show that a human EGFR mutant lung cancer cell line, such as 1975, doesn't cause MPE. This is a more pertinent control to A549 than an EGFR transfected cell. Since CCL2 is typically driven by EGFR mutation, it would be surprising if they don't make MPE.

3. Use of CD11b+/Gr1+ as MDSC definition. Given the importance placed specifically on the monocyte lineage here (CCL2), the CD11b+/Gr1+ markers are inadequate. For the most part, these markers have been replaced with CD11b+/Ly6C+/Ly6G- for monocytes and

CD11b/Ly6C+/Ly6G+ for neutrophils. The majority of Gr1+ cells in any model system will be neutrophils. The authors should specifically track monocyte and macrophages. Are these cells actually present in the pleural space?

4. Mast cell experiments. These experiments are intriguing, but raise many mechanistic questions that have not been addressed. Specifically, how does the presence of mast cells tie into this phenotype? The authors show that there aren't any MDSC in the spleen in mast cell deficient mice. Why is that? Is this a general function of mast cells or specific to MPE? Since most tumor models in mice lead to accumulation of MDSC in spleens, mast cells could be important for this process in the absence of MPE. A simple flank tumor allograft with LLC may uncover this. Again, the mechanisms that may explain this have not been addressed. Since the mast cell experiments are not essential to this story, the authors may choose to delete these experiments. If they remain, additional mechanistic detail is needed.

5. Is the process slowed down in splenectomized mice or eliminated? A later time point may answer this question.

6. Minor. CCL3 and CCL4 usually track with CCL2 and could also perform similar functions. It would be useful to see if they are also elevated in this context.

RESPONSE TO REVIEWERS' COMMENTS

Reviewer #1

“A. In this manuscript the authors consider the relationship between tumor expression of KRAS and the development of malignant pleural effusion (MPE). They also examine the role of CCL2, CD11b+/Gr1+ cells and mast cells in this process. They observe that KRAS positive tumors are more likely to induce MPE, that reduced KRAS expression is associated with reduced MPE and that MPE occurs by a mechanism associated with tumor cell expression of CCL2, mast cells and the recruitment of CD11b+/Gr1+ positive cells. Notably, pharmacological KRAS inhibition reduces MPE.

B. MPE is a serious clinical problem that requires attention. The role of KRAS has not previously been examined in this setting. However the role of mast cells and the predominance of CD11b+/Gr1+ cells in the infiltrate has previously been observed, as referenced by the authors.

C. The tumour models chosen for study were an assortment of different cell types from a variety of tissue sources, several of those that induced MPE were of an airways epithelial origin (eg LLC and A549) a more systematic examination of specific types of tumor cells would have been more convincing. There are many databases of tumour profiles and clinical information which could have been used to determine if the association between KRAS and MPE held true on a larger clinical scale.”

We thank the Reviewer for her/his comprehensive evaluation of our manuscript and for recognizing the clinical problem at stake. This Reviewer is right in stating that “The tumour models chosen for study were an assortment of different cell types from a variety of tissue sources”. This is also true for clinical MPE that is caused by virtually any tumour type. This is why we chose cell lines of different tissue sources; i.e. because they need to have a common effector to cause MPE. And we discovered it: it is KRAS mutations that cause CCL2 overexpression and lead to MPE. Also, the cell lines used were the only ones available to us, since there are not many mouse cell lines out there that are syngeneic to the *C57BL/6* mouse. Actually, the models of MPE reported herein are probably all of the existing ones and were developed by us for the purposes of this paper. We have now included clinical data that strengthen the link between mutant *KRAS* and malignant pleural effusion. We have also included data from an isogenic cellular system of primary chemical-induced lung adenocarcinoma cell lines that supports a role for *KRAS* in MPE.

“The mast cell dependency of neutrophil and in some cases monocyte recruitment has been demonstrated in many situations. However, the use of a single type of mast cell deficient mouse together with a reconstitution protocol which, uses a very high dose of mast cells and does not allow the pleural mast cells to differentiate into the

type of mast cell found locally in this setting before responding to tumours, limits the interpretation of the data provided.”

We have previously shown (ref. 7 from main paper) that bone marrow-derived mast cells are recruited to the pleural space upon MPE induction and that their local numbers cannot be explained by the local tissue depots. In the above paper, we also showed that this number of mast cells results in pleural transmigration of a number of mast cells that is similar to what is actually found in mouse MPE. However, the Reviewer is absolutely correct: mast cell dependency of neutrophil and in some cases monocyte recruitment has been demonstrated in many situations and is not our main point. To alleviate the Reviewer’s critique, we removed the data relevant to mast cells from the revised version of the manuscript, as mast cells were not a focus of this work and the methods employed were non-physiologically relevant.

“CD11b+/Gr1+ cells will include a number of cells types, likely mostly neutrophils in this setting, only a proportion of which will have MDSC characteristics. The interpretation that these cells are mainly MDSC is not justified. Further markers and/or functional data would be required. The quality of the single micrograph showing MPE associated infiltrating cells is also very poor. A better analysis of the proportions of neutrophils, monocytes and eosinophils in the MPE would be informative.”

The Reviewer is absolutely correct. We have now removed unsubstantiated terms such as “myeloid-derived suppressor cells”, which were used off-target in the first place. We simply refer to freshly recruited bone marrow cells that express CD11b+ and Gr1+. We do not wish to confer nor imply immunosuppressant functions to these cells. MPE development triggered by KRAS-mutant cancer cells was associated with an influx of increased numbers of myeloid cells into the pleural space, but not with the presence of newly-appearing morphologically distinct cell types, since differential pleural cell counts were similar in mice with or without a MPE. These issues were fixed throughout the text.

“Given the impact of KRAS on metabolic activities of tumour cells, it may be important to distinguish between the ability of KRAS to allow tumours to thrive in a low oxygen environment where they promote MPE from any CCL2 production associated impact of KRAS on MPE induction. Use of CCR2 deficient animals, as hosts, could help address this question.”

This is a well taken and pertinent comment. Hypoxia could shape the environment responsible for the different behavior of KRAS-mutant tumour cells that have metastasized to the pleural space. However, we have no way of presently addressing this experimentally, but mentioned this point in the revised discussion section.

“D. More details should be provided on the statistical approach used in each figure. In some cases, it is unclear if experiments were replicated or if the results presented are from a single experiment using several mice. In other situations the statistical interpretation does not seem to match the data as presented. For example in the final panel of figure 6D (1st bar vs 3rd bar and 2nd bar vs 3rd bar).”

Statistics were made explicit throughout the text, figures, and legends thereof.

“E. With the data provided, while the conclusion can be drawn that KRAS expression enhances MPE in some animal tumour models it is unclear how this is occurring and whether this is a feature of human disease.”

We have now included clinical data that strengthen the link between mutant *KRAS* and malignant pleural effusion and that show scenarios of how mutant *KRAS* can go undetected in clinical samples. The pertinent point of this Reviewer was made in the revised discussion section.

“G. References are appropriate”

We thank this Reviewer for the kind comment.

“H. Some over interpretation of data from murine models. Especially with respect to role of mast cells CCL2 and MDSC. In order to suggest this is a feature of clinical disease, need much more data from a systematic study of human tumors. The discussion is overly long and complex. The figure legends do not give sufficient information.”

These issues were corrected in the revised version of the manuscript.

Reviewer #2

“Dear Authors,

This is an elegant series of well conducted experiments reviewing the mechanism behind MPE formation in a mouse model of MPE, including the use of human tumour cell lines. The experiments appear to show that KRAS mutation is both necessary and sufficient for MPE formation, and that this acts through a MC dependant manner, requiring splenic trafficking. MPE formation appears blocked by a deltarasin. This is a highly novel area of substantial clinical interest. The methods used appear sound and the data is largely clearly presented (I think too many figures are presented and the main figures only should be presented in the main manuscript, maximum of 5). Conclusions made by the authors appear largely valid, however, I have several areas of comment:"

We thank this Reviewer for his/her efforts and generous comments. We have abided by his/her criticism and have now reduced the number of Figures to 5, while moving everything else online.

"1. As above, there are too many figures and the authors should choose the most relevant 5 figures and present these only with the others moved to online."

We have now reduced the number of Figures to 5, while moving everything else online.

"2. A key issue (which the authors address) is that currently in human treatment, KRAS mutation is not commonly found. Some rather weak arguments are presented as to why this should be, but the authors experiments appear to show that KRAS mutation is absolutely required for MPE formation - the other potential conclusion (in addition to the authors which is that our clinical assays for KRAS are not yet sensitive enough) is that this pathway is only relevant in the presented mouse model, and this limitations should be discussed. If KRAS was so essential in human MPE, we would expect to see different KRAS results from biopsy."

We have now included clinical data that strengthen the link between mutant *KRAS* and malignant pleural effusion and that show scenarios of how mutant *KRAS* can go undetected in clinical samples. The pertinent point of this Reviewer was made in the revised discussion section.

"3. While patients with and without pleural effusion from malignancy exist, the vast majority (I would estimate 95%) are associated with effusion. This again suggests that some form of KRAS mutation should be detected in nearly all MPE / malignant pleural patients and the authors should discuss this."

We agree with the Reviewer 100%. We do not claim that KRAS mutations are the only ones that cause MPE in humans and postulate mutant KRAS effects to be class effects shared by all driver mutations aligned along the KRAS pathway, including EGFR, KRAS, PIK3CA, BRAF, MET, EML4/ALK, RET, and other mutations. To this end, mutant EGFR was recently shown to cause MPE when expressed in H1299 human lung adenocarcinoma cells (ref 17 from main text). However, possible pathogenic roles for other lung cancer drivers in MPE remain to be shown. Together with the advent of MPE sequencing techniques, such developments could lead to targeted therapies for MPE in the near future. We discuss this point in the revised discussion section.

“4. MPE is a clinically heterogenous set of diseases from a number of primary sites - this may mean that some / all of the pathway delineated is relevant to some patients and not others.”

Again, we agree with the Reviewer. The pathway examined here is more relevant to lung, pancreatic, and colon cancer, while in other tumors (ie, breast cancer) other mutations may be functionally involved in MPE formation (ie, HER2). This pertinent point is now discussed in the revised discussion section.

“5. The last figure concludes beyond the data and is speculative - please remove completely.”

The schematic was completely removed.

“6. I would encourage the authors to conclude within the data and suggest further steps necessary in human MPE to prove their theories.”

We included potential human study design to prove this hypothesis in humans in the revised discussion section and thank again the Reviewer for the pertinent comment.

Reviewer #3

“The authors found KRAS-dependent CCL2/CCR2 paracrine signaling, resulting in recruitment of bone marrow derived MC and subsequently of splenic residing CD11b+Gr1+ MDSC into the tumor afflicted pleural space and accumulation of pleural effusion. These are new and interesting findings from a series of cell lines and in vivo animal studies.”

We thank this Reviewer for his/her efforts and generous comments.

“1. The animal numbers of each group are quite variable, such as n=8-27/group in Figure 7. This is not the standard of experimental design. Could the authors describe what animal numbers of each group in the design are?”

The animal numbers are variable because experiments were done in a fractionated fashion and not all at once, until statistical significance was reached. We usually started with *n* close to 10, and added controls and intervention animals as needed. Because no outliers were excluded from analyses, more animals were added to follow-up experiments in the groups where outliers spread the dispersion of the data. This is the reason for the deviant animal numbers in some *in vivo* experiments. The numbers of animals used are now described in detail in the figure legends. The experimental design is now better delineated in the Methods.

“2. MDSC play an important role in immunosuppression in tumor microenvironment. What is the mechanism that CD11b+Gr1+ MDSC recruited into plural space causes inflammation and angiogenesis shown in the Supplementary Figure 1?”

The Reviewer is absolutely correct in that the term MDSC does not apply to the cells we refer to. We have now removed unsubstantiated terms such as “myeloid-derived suppressor cells”, which were used off-target in the first place. We simply refer to freshly recruited bone marrow cells that express CD11b+ and Gr1+. We do not wish to confer nor imply immunosuppressant functions to these cells. We simply show in new Extended Data Fig. 7 (which was previously presented in the main article figures) that these cells contribute to MPE development and are probably educated *in vivo* by KRAS mutant tumor cells, likely via systemic CCL2 shed. These issues were fixed throughout the text.

“3. The authors concluded that mEGFR is neither sufficient nor required for experimental MPE induction by the results of Figures S4A and S4B. This is quite contradictory to clinical observation that MPEs occur in every molecular subtypes (K-Ras mutation, EGFR mutation, ALK translocation, ROS-1 translocation, etc).. and histological subtypes (small cell lung cancer, adenocarcinoma, squamous cell carcinoma) of lung cancers. The authors comment in the discussion that mutant KRAS bearing tumors cells causing MPE would escape detection in focal pleural tumor tissue biopsies and low-volume pleural fluid aspirates. This is quite difficult to understand in the genomic era that sensitive methods (esp NGS) been used to find the driver genes for various targeted therapies, which are listed in guidelines of many organizations. In addition, the authors used mEGFR overexpression in B16F10 cell to

show no MPE formation. The EGFR protein expression of the B16F10pΔEGFR was quite low in supplementary figure 2g. The experiment that knowndown of EGFR in mutant KRAS cell does not affect the activity of KRAS nor decrease the amount MPE can't be the evidence that mutant EGFR does not from MPE.”

We agree with the Reviewer 100%. We do not claim that *KRAS* mutations are the only ones that cause MPE in humans and postulate mutant *KRAS* effects to be class effects shared by all driver mutations aligned along the *KRAS* pathway, including *EGFR*, *KRAS*, *PIK3CA*, *BRAF*, *MET*, *EML4/ALK*, *RET*, and other mutations. To this end, mutant *EGFR* was recently shown to cause MPE when expressed in H1299 human lung adenocarcinoma cells (ref 17 from main text). However, possible pathogenic roles for other lung cancer drivers in MPE remain to be shown. Together with the advent of MPE sequencing techniques, such developments could lead to targeted therapies for MPE in the near future. We discuss this point in the revised discussion section. The main focus of this manuscript is the role of *KRAS* in MPE. We do not wish nor can we prove based on the weak data presented in the original version of the manuscript to discredit *EGFR* signaling as a possible cause of MPE. We have thus amended ambiguous data that are off-target from the focus of this work. We have removed every reference to the functional role of *EGFR* in MPE, as this was not a focus of this work and the data were ambiguous. We have also reformatted the discussion to accommodate the Reviewer's valid concern. The pathway examined here is more relevant to lung, pancreatic, and colon cancer, while in other tumors (ie, breast cancer) other mutations may be functionally involved in MPE formation (ie, *HER2*). This pertinent point is now discussed in the revised discussion section.

4. The supplementary figure S3c does not support the statement -"manipulation of KRAS signaling did not result in obvious change in tumor cell proliferation" in the Results.”

This valid point was corrected and the relevant phrase of the results section was corrected to: “Manipulation of *KRAS* signaling did not result in obvious enhancements of tumour cell proliferation or survival in vitro; on the contrary, overexpression of *KrasG12C2A* in *PANO2* cells and of *KrasG12C2B* in *B16F10* cells slowed their growth rate (Extended Data Figure 4)”.

Reviewer #4

“The authors use a large series of mouse based experiments to show essential roles for mutant *K-ras*, *CCL2*, *CD11b+/Gr1+* cells, and mast cells in the development of malignant pleural effusion (MPE). There is an impressive number of cell lines employed. The data is of high quality and appears quite reproducible. A series of experiments using splenectomy models, bone marrow tagged cells, *Ccr2-/-* and mast

cell-deficient mice are provided to uncover the operative mechanisms at play. In summary, this study includes a large amount of data to support the main conclusions and is quite novel. However, there are some deficiencies in the study that must be addressed.”

We deeply appreciate the efforts spent by the Reviewer in order to improve our manuscript. We have undertaken every effort feasible to address the concerns raised.

“1. Lack of human correlation. The authors attempt to explain that this wasn't done because it would be overly difficult. No human study is straight forward, but this would be far from difficult to conduct. At most centers in the US, all stage IV lung cancer patients undergo genetic testing for EGFR and ALK. The remaining DNA could be used to genotype KRAS. Since these are all stage IV patients, many of them will have MPE. It should not be difficult to establish MPE rates for KRAS mutant vs. EGFR mutant vs WT for both. The way the current study is written suggests that KRAS is required for MPE. It is difficult to imagine that all MPE is related to KRAS in human disease.”

We have now included clinical data that strengthen the link between mutant *KRAS* and malignant pleural effusion and that show that *KRAS* mutations are related with pleural metastasis, are frequent in MPE, and also include scenarios of how mutant *KRAS* can go undetected in clinical samples. The pertinent point of this Reviewer was made in the revised discussion section.

“2. Inclusion of an EGFR mutant cell line. The authors do a great job of including numerous cell lines of both mouse and human origin. One concern is that they are of many different histologies. It is important to show that a human EGFR mutant lung cancer cell line, such as 1975, doesn't cause MPE. This is a more pertinent control to A549 than an EGFR transfected cell. Since CCL2 is typically driven by EGFR mutation, it would be surprising if they don't make MPE.”

The main focus of this manuscript is the role of *KRAS* in MPE. We do not wish nor can we prove based on the weak data presented in the original version of the manuscript to discredit EGFR signaling as a possible cause of MPE. We have thus amended ambiguous data that are off-target from the focus of this work. We have removed every reference to the functional role of EGFR in MPE, as this was not a focus of this work and the data were ambiguous. We have also reformatted the discussion to accommodate the Reviewer's valid concern. In summary, we do not claim that *KRAS* mutations are the only ones that cause MPE and postulate mutant *KRAS* effects to be class effects shared by all driver mutations aligned along the *KRAS* pathway, including EGFR, *KRAS*, PIK3CA, BRAF, MET, EML4/ALK, RET, and other mutations. The pathway examined here is more relevant to lung, pancreatic, and

colon cancer, while in other tumors (ie, breast cancer) other mutations may be functionally involved in MPE formation (ie, HER2). However, possible pathogenic roles for other lung cancer drivers in MPE remain to be shown. Together with the advent of MPE sequencing techniques, such developments could lead to targeted therapies for MPE in the near future. We discuss this point in the revised discussion section.

“3. Use of CD11b+/Gr1+ as MDSC definition. Given the importance placed specifically on the monocyte lineage here (CCL2), the CD11b+/Gr1+ markers are inadequate. For the most part, these markers have been replaced with CD11b+/Ly6C+/Ly6G- for monocytes and CD11b/Ly6C+/Ly6G+ for neutrophils. The majority of Gr1+ cells in any model system will be neutrophils. The authors should specifically track monocyte and macrophages. Are these cells actually present in the pleural space?”

Yes, all of these cells are present in experimental MPE in millions and have been described previously both morphologically and molecularly in previous related publications (J Clin Invest. 2015 Jun;125(6):2317-34. PLoS One. 2013 Aug 14;8(8):e71207. Am J Respir Crit Care Med. 2012 Sep 15;186(6):487-92. Oncogene. 2013 Jan 24;32(4):528-35.

Am J Respir Crit Care Med. 2010 Nov 15;182(10):1273-81. Mol Cancer. 2010 Mar 10;9:56. J Natl Cancer Inst. 2008 Oct 15;100(20):1464-76. Am J Respir Crit Care Med. 2008 Jul 1;178(1):50-9. Cancer Res. 2007 Oct 15;67(20):9825-34. Am J Respir Cell Mol Biol. 2006 Feb;34(2):142-50). The Reviewer is absolutely correct. We have now removed unsubstantiated terms such as “myeloid-derived suppressor cells”, which were used off-target in the first place. We simply refer to freshly recruited bone marrow cells that express CD11b+ and Gr1+. We do not wish to confer nor imply immunosuppressant functions to these cells. We have also used Ly6c and Ly6g in addition to CD11b and Gr1 and always come to the same conclusion. MPE development triggered by KRAS-mutant cancer cells was associated with an influx of increased numbers of myeloid cells of all kinds into the pleural space (both CD11b+Gr1+Ly6c+ and CD11b+Gr1+Ly6g+), but not with the presence of newly-appearing morphologically or molecularly distinct cell types, since differential pleural cell counts and flow cytometry results were similar percentagewise in mice with or without a MPE. The difference is absolute numbers, as also shown real time by our bioluminescent chimera approach: Half a million myeloid cells are present in the normal pleural space and in mice with dry pleural tumors, but several million of these cells stream in during MPE development. These issues were addressed throughout the text.

“4. Mast cell experiments. These experiments are intriguing, but raise many mechanistic questions that have not been addressed. Specifically, how does the presence of mast cells tie into this phenotype? The authors show that there aren't any MDSC in the spleen in mast cell deficient mice. Why is that? Is this a general function of mast cells or specific to MPE? Since most tumor models in mice lead to accumulation of MDSC in spleens, mast cells could be important for this process in the absence of MPE. A simple flank tumor allograft with LLC may uncover this. Again, the mechanisms that may explain this have not been addressed. Since the mast cell

experiments are not essential to this story, the authors may choose to delete these experiments. If they remain, additional mechanistic detail is needed.”

We concur with the Reviewer in that the mast cell experiments are not critical to this work and would require additional mechanistic studies that would not add anything to the present story. Hence we removed all mast cell-related data to alleviate this Reviewer’s valid criticism.

“5. Is the process slowed down in splenectomized mice or eliminated? A later time point may answer this question.”

Splenectomy-conferred protection was long-lived, as even mice harvested 30 days post-tumour cells did not have MPE, so we did late time points and the mice still did not get MPE at 4 weeks while controls had MPE at 2 weeks (Extended Data Fig. 7c). We have not followed-up mice longer, but this prolonged survival induced by splenectomy we believe is clinically relevant.

“6. Minor. CCL3 and CCL4 usually track with CCL2 and could also perform similar functions. It would be useful to see if they are also elevated in this context.”

CCL3 and CCL4 were not overexpressed by KRAS-mutant tumour cells compared with wild-type ones in our microarrays, which are available for the Reviewer online, so we did not pursue those chemokines.

Reviewers' comments:

Reviewer #1 (Remarks to the Author):

The authors have responded to many of the comments made by simply removing the relevant data or comments (e.g. comments related to role of mast cells and identification of MDSC). This has somewhat improved the focus of the paper but also reduces the overall impact and mechanistic depth of the report. This is offset to some extent by the additional clinical information provided.

There remain concerns regarding the limited choice of lines used in the C57Bl/6 model. Several other cell lines are often used in this strain of mice e.g. E0771, TRAMP-C1 cells, MC57G but have been excluded from the current analysis. To support the author's conclusions a wider range of tumors should be examined. They are relying on very few tumors as a basis

The link between mutant KRAS and enhanced tumor growth is also not very carefully considered in the in vivo context. Some further information on the overall tumor burden and its relationship to MPE would greatly strengthen the author's conclusions. This is also a significant issue when considering the impact of daltarasin in vivo, where reduced overall tumor growth rates would be predicted as a result of treatment in mutant KRAS expressing cells.

A particular concern was the authors reply to comments from a reviewer regarding the varying "n" numbers used for different experiments. It appears from their reply that additional animals were added to certain experimental groups (but not others) until statistical significance was reached. This is clearly not good scientific practice and places doubt on the validity of findings in a number of places in the manuscript. For example, CULA data from Figure 1A; Deltarasin treatment group data in Figure 5E.

Reviewer #2 (Remarks to the Author):

Dear Authors,

Many thanks for addressing some of my concerns in your revision. There is still a theme along the whole manuscript that KRAS mutation is the only MPE driving mutation of interest, and several conclusions should be softened to reflect this. The diverse nature of human MPE should be discussed further, with the important results shown discussed in this context as applicable to the variety seen in clinical MPE practice in humans.

Reviewer #3 (Remarks to the Author):

In this revised manuscript, the authors addressed clearly almost all of the points raised by the reviewers. There is only one minor point that needs to be clarified. In Figure 4d, the numbers of lung cancer patients in each group (WT, EGFR, KRAS) should be shown to make it clear what is the percentage of metastatic sites.

Reviewer #4 (Remarks to the Author):

No additional comments.

RESPONSE TO REVIEWERS' COMMENTS

Reviewer #1 (Remarks to the Author):

There remain concerns regarding the limited choice of lines used in the C57Bl/6 model. Several other cell lines are often used in this strain of mice e.g. E0771, TRAMP-C1 cells, MC57G but have been excluded from the current analysis. To support the author's conclusions as wider range of tumors should be examined. They are relying on very few tumors as a basis.

We want to thank the Reviewer for the in-depth analysis of our manuscript and the constructive criticism. We have done everything within our powers to address the concern raised. In detail, we have now performed statistical analysis of the number of cell lines used that indicates beyond doubt that the sample we assessed is sufficient to prove our hypothesis. We cannot possibly include new cell lines in this dataset as requested, since we cannot justify it to our local IACUC authorities, as we have proven our hypothesis in any feasible way. In brief, we have tested 12 cell lines and have generated 18 daughter cell lines with modulated KRAS expression. In summary, out of the 12 parental cell lines tested, six out of six *KRAS*-mutant cell lines were MPE-competent and none MPE-incompetent, while none out of six *KRAS*-wild-type cell lines was MPE-competent, and six MPE-incompetent ($P = 0.0022$; χ^2 test), indicating a statistically significant association between mutant *KRAS* and MPE. *KRAS* expression modulation via shRNA or plasmid-mediated expression of mutant *KRAS* also inhibited or induced MPE formation, respectively, in all 10 daughter cell lines used ($P < 0.0001$; χ^2 test), indicating that *KRAS* is important in MPE development in mice.

“The link between mutant *KRAS* and enhanced tumor growth is also not very carefully considered in the in vivo context. Some further information on the overall tumor burden and its relationship to MPE would greatly strengthen the author's conclusions. This is also a significant issue when considering the impact of deltarasin in vivo, where reduced overall tumor growth rates would be predicted as a result of treatment in mutant *KRAS* expressing cells.”

As requested by the Reviewer, we also included new data of pleural tumor burden in new Figures 1 and 2 that indicate that pleural tumour growth is NOT associated with MPE development. This is also a well-known clinical pearl: patients with massive pleural tumour burden may be “dry”, whereas patients with minimal pleural tumour involvement may have massive amounts of pleural fluid (). This is also true for experimental models (Hatton MW, Southward SM, Ross BL, Clarke BJ, Singh G, Richardson M. Relationships among tumor burden, tumor size, and the changing concentrations of fibrin degradation products and fibrinolytic factors in the pleural effusions of rabbits with VX2 lung tumors. *J Lab Clin Med.* 2006 Jan;147(1):27-35). We did measure pleural tumor mass in the first experiments shown

in Figures 1 and 2 and show now the data. Since this was not correlated with MPE volume, we stopped pursuing this end-point.

A particular concern was the authors reply to comments from a reviewer regarding the varying “n” numbers used for different experiments. It appears from their reply that additional animals were added to certain experimental groups (but not others) until statistical significance was reached. This is clearly not good scientific practice and places doubt on the validity of findings in a number of places in the manuscript. For example, CULA data from Figure 1A; Deltarasin treatment group data in Figure 5E.

We believe that our experimental approach was misunderstood. All of the experiments shown and conducted in our laboratory are well designed and controlled. What we meant is that, when our power analysis tells us that we need $n = 15$ per group, we always do a first experiment with all the groups with $n = 7-10$ followed by interim analyses and one or multiple follow-up experiments. In these follow-up experiments, the case:control ratio may be changed from 1:1 to 1:2 or even 1:3 in favor of the control or intervention groups, as per interim power analyses. This way we can rule in or out statistical significance with fewer mice and we believe it is excellent scientific practice. For example, in Figure 5e, 27, 10, and 20 mice are shown in the three groups. In this experiment, the case:control ratio was 1:1, i.e. each deltarasin and each CCL2 antibody-treated mouse had its control and mice from all three groups were injected in parallel in two separate batches. Three mice from the control group died before harvest. We believe that this way of conducting the experiments (in fractionated batches of control and matched interventional mice followed by interim analyses) is well-designed to reach or fail statistical significance with the least number of mice. We now added more data (n) to some groups from Figure 1 to accommodate this Reviewers' concerns of variable n . Again, all groups of mice were injected in parallel with other groups of mice, i.e. all experiments were controlled.

Reviewer #2 / Expert in Malignant pleural effusion

“Reviewer #2 (Remarks to the Author):

Dear Authors,

Many thanks for addressing some of my concerns in your revision. There is still a theme along the whole manuscript that KRAS mutation is the only MPE driving mutation of interest, and several conclusions should be softened to reflect this. The

diverse nature of human MPE should be discussed further, with the important results shown discussed in this context as applicable to the variety seen in clinical MPE practice in humans.”

We agree with this Reviewer’s pertinent comment and have now softened every hard or misleading expression that might misleadingly convey that KRAS mutations are solely responsible for MPE throughout the text and even the title. This pertinent point is discussed in the revised Discussion section.

Reviewer #3 / Expert in Lung Cancer

“Reviewer #3 (Remarks to the Author):

In this revised manuscript, the authors addressed clearly almost all of the points raised by the reviewers. There is only one minor point that needs to be clarified. In Figure 4d, the numbers of lung cancer patients in each group (WT, EGFR, KRAS) should be shown to make it clear what is the percentage of metastatic sites.”

We thank this Reviewer for his/her efforts and generous comments. We have revised former Fig. 4, now new Table 4 to address the concern of this Reviewer including the percentage of patients in each category.

Reviewer #4 / Expert in Lung Cancer microenvironment and Immunology

“Reviewer #4 (Remarks to the Author): No additional comments.”

We deeply appreciate the efforts spent by the Reviewer in order to improve our manuscript.

REVIEWERS' COMMENTS:

Reviewer #1 (Remarks to the Author):

No further comments

Reviewer #2 (Remarks to the Author):

I have no further comments